# PARAMETER-EFFICIENT MULTI-SOURCE DOMAIN-ADAPTIVE PROMPT TUNING FOR OPEN-VOCABULARY OBJECT DETECTION

## ABSTRACT

Cross-domain open-vocabulary object detection (COVD) poses a unique and underexplored challenge, requiring models to generalize across both domain shifts and category shifts. To tackle this, we propose MAP: a parameter-efficient Multi-source domain-Adaptive Prompt tuning framework that leverages multiple labeled source domains to improve detection in novel, unlabeled target domains with unseen categories. MAP consists of two key components: Multi-Source Prompt Learning (MSPL) and Unsupervised Target Prompt Learning (UTPL). MSPL disentangles domain-invariant category semantics from domain-specific visual patterns by jointly learning shared and domain-aware prompts. UTPL enhances generalization in the unlabeled target domain by enforcing prediction consistency under text-guided style augmentations, introducing a novel entropy-minimization objective without relying on pseudo-labels. Together, these components enable effective alignment of visual and textual representations across both domains and categories. In addition, we present a theoretical analysis of the proposed prompts, examining their behavior through the lenses of fidelity and distinction. Extensive experiments on challenging COVD benchmarks demonstrate that MAP achieves state-of-the-art performance with significantly fewer additional parameters.

## 1 INTRODUCTION

Object detection has witnessed significant progress with the emergence of large-scale vision-language models (VLMs), which enable open-vocabulary detection (OVD) by leveraging textual supervision to recognize a wide range of categories, including those unseen during training Phoo & Hariharan (2022); Lin et al. (2022); Feng et al. (2022); Yao et al. (2022); Minderer et al. (2022); Zareian et al. (2021); Zhou et al. (2022c); Arandjelović et al. (2023); Cheng et al. (2024). However, most existing OVD methods assume that training and test data share the same underlying domain, which is often unrealistic in practical scenarios where data distribution shifts are common. This motivates the task of Cross-domain Open-Vocabulary Object Detection (COVD), where models must generalize to unseen categories in a novel domain.

Prompt learning has recently emerged as a powerful and flexible strategy for adapting VLMs to downstream tasks. Instead of fine-tuning the entire model, prompt learning optimizes a small set of learnable tokens that guide the pre-trained model to focus on task-relevant features. This parameter-efficient approach has shown strong performance in domain adaptation Wang et al. (2024b); Singha et al. (2023; 2024); Li et al. (2024); Ge et al. (2023). In the context of COVD where models must handle both domain and category shifts, prompt learning offers a natural and scalable solution, as it enables the model to flexibly align textual and visual representations across diverse distributions. In this work, we introduce Multi-Source Prompt Learning (MSPL) for the first time in COVD, leveraging multiple labeled source domains to jointly capture class-specific and domain-specific knowledge. By learning both domain-adaptive and class-aware prompts, MSPL enables effective visual-textual alignment across domains and categories, and supports robust generalization to unseen target domains with novel classes.

For target adaptation, previous methods rely heavily on pseudo labels generated from source-trained models Chen et al. (2023); Yang et al. (2024a;b); Liu et al. (2025); Wang et al. (2024a). Such pseudo labels are often unreliable due to domain and category shifts, especially in open-vocabulary settings

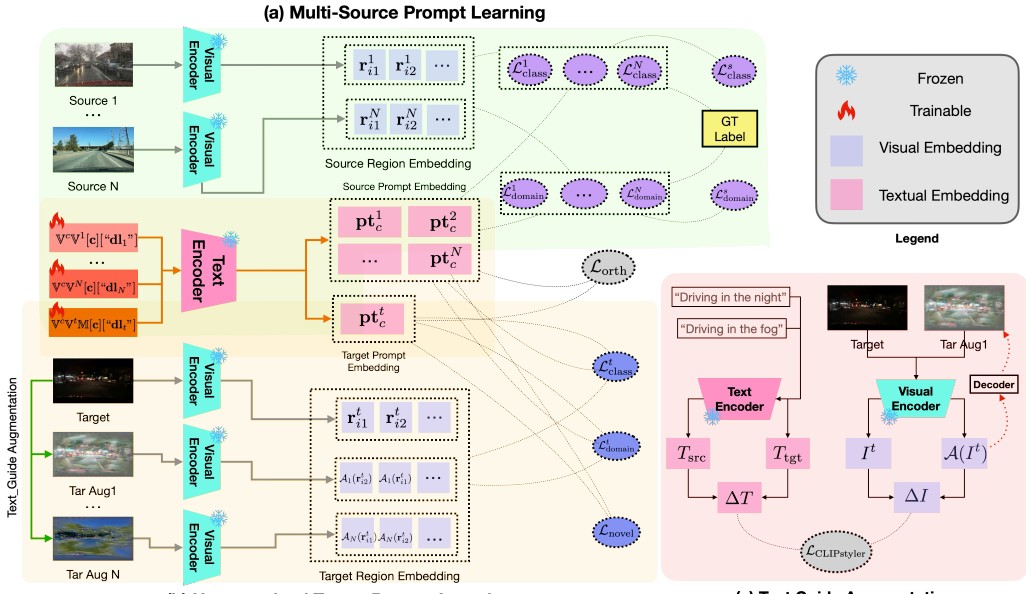

Figure 1: Framework overview: (a) Multi-Source Prompt Learning (MSPL) disentangles domain-invariant category semantics from domain-specific visual patterns by jointly learning shared and domain-aware prompts. (b) Unsupervised Target Prompt Learning (UTPL) enhances generalization in the unlabeled target domain by enforcing prediction consistency under text-guided style augmentations (c), introducing a novel entropy-minimization objective without relying on pseudo-labels. See Appendix C for detailed loss illustration.

where unseen classes lack meaningful supervision. This limitation hinders robust generalization and leads to error propagation. To address these challenges, we propose Unsupervised Target Prompt Learning (UTPL), which eliminates the need for pseudo labels by enforcing prediction consistency under style-guided augmentations through a novel entropy-based objective. Together with MSPL, UTPL forms the core of our framework, Multi-source domain-Adaptive Prompt tuning (MAP), which enables robust adaptation to novel domains and categories in the COVD setting.

Beyond the algorithmic design, we provide a theoretical analysis of the learnable prompts in MAP by introducing two key properties: fidelity and distinction, to characterize their effectiveness. We rigorously define these properties through an information-theoretic lens, connecting high-fidelity prompts to low predictive entropy and high mutual information with latent labels, and high-distinction prompts to mutual orthogonality across domains or classes. Furthermore, we propose a novel definition of unsupervised fidelity, enabling the evaluation of prompt quality even in the absence of labeled target data. This theoretical grounding helps to clarify why MAP is effective and provides a principled framework for future developments in prompt-based domain adaptation.

Our main contributions are as follows: (1) We propose MAP, a novel framework for multi-source cross-domain open-vocabulary object detection that integrates MSPL and UTPL to align visual–textual representations across domains and categories. (2) We provide a theoretical analysis of prompt fidelity and distinction, introducing a new formulation of unsupervised fidelity based on entropy and mutual information. (3) We achieve state-of-the-art performance on standard COVD benchmarks with minimal additional parameters.

## 2 RELATED WORKS

**Open vocabulary object detection.** Open-vocabulary detection was first proposed by Bansal et al. Bansal et al. (2018), introducing the visual-semantic embeddings to replace the classification layer of a closed vocabulary detector. This design has now become a common practice for many subsequent open-vocabulary object detection works Phoo & Hariharan (2022); Lin et al. (2022); Feng et al. (2022); Yao et al. (2022); Minderer et al. (2022); Zareian et al. (2021); Zhou et al. (2022c); Arandjelović et al. (2023); Cheng et al. (2024). To name a few, ViLD Phoo & Hariharan (2022)

uses a teacher-student framework to distill knowledge from a pre-trained open-vocabulary image classification model (i.e., teacher model) and detect region boxes with the student model by aligning the text and image embeddings. PromptDet Feng et al. (2022) scales up the OV detector with noisy uncurated web images using the proposed regional prompt learning technique which aligns text embedding with visual features. DetClip Yao et al. (2022) designs a concept dictionary with a paralleled concept formulation to facilitate learning from heterogeneous datasets including detection, grounding, and image-text pairs. YOLO-World Cheng et al. (2024) is built upon the YOLO backbone and highly generalizable vision-language models; a vision-language path aggregation network is further utilized to enhance the interaction between visual and semantic information. In this work, we use several pre-trained open-vocabulary object detectors as backbones to evaluate their ability over cross domain adaptation.

**Domain adaptive prompt learning.** Prompting origins in NLP tasks which prepend instructions to the input to help train the language model Lester et al. (2021); Shin et al. (2020). With the emergence of vision-language models such as CLIP Radford et al. (2021), ALIGN Jia et al. (2021), which embed a shared visual-semantic space by learning from a large scale of image alt-text pairs using a contrastive loss, domain-adaptive prompt learning has been used to address domain shifts between the pre-trained VLMs and the downstream tasks Ge et al. (2023); Li et al. (2024); Chen et al. (2024); Bai et al. (2024); Zhao et al. (2024); Wang et al. (2024b). One line of the work constructs domain adaptive prompts by directly embedding domain information Ge et al. (2023); Li et al. (2024); Bai et al. (2024). The other line of work aims to align source and target domains with a prompt-based framework Chen et al. (2024); Cao et al. (2024); Bai et al. (2024). To facilitate multi-source adaptive prompt learning (MSPL), MPA Chen et al. (2023) first trains individual prompt for a source and target domain pair with a contrastive loss then utilizes auto-encoding to denoise the learned prompt and align all the reconstructed prompts. SAP-CLIP Yang et al. (2024a) designs learnable prompts with class context vectors and domain representation vectors based on CLIP to leverage textual information of both class semantics and domain representations. APNE-CLIP utilizes negative textual semantics and energy-based uncertainty to transfer knowledge from multiple source domains Yang et al. (2024b). VAMP proposes a vision-aware multimodal prompt tuning framework to address multi-source domain few-shot adaptation Liu et al. (2025). However, these multi-source prompt learning methods heavily rely on pseudo labels generated by source-trained detectors, which are often noisy and unreliable—especially under significant domain and category shifts. In this work, we propose MAP (Multi-source domain-Adaptive Prompt tuning) that circumvents the reliance on pseudo labels by introducing a novel unsupervised target prompt learning (UTPL) strategy.

## 3 METHODOLOGY

**CLIP.** The CLIP model Radford et al. (2021) consists of a visual encoder $f$ and a text encoder $g$ that map images and text into a shared embedding space. To enable zero-shot classification, class labels are converted into textual prompts, e.g., "a photo of a [CLASS]". After tokenization, the prompt $\mathbf{pt}$ is encoded into a text embedding $\mathbf{z} = g(\mathbf{pt})$. Given an image $\mathbf{x}$ with label $y$, the visual encoder produces an embedding $f(\mathbf{x})$. The probability of class $k$ is then computed using cosine similarity between the visual and text embeddings:

$$p(\hat{y} = k|\mathbf{x}) = \frac{\exp(\cos(f(\mathbf{x}), g(\mathbf{pt}k))/T)}{\sum c = 1^K \exp(\cos(f(\mathbf{x}), g(\mathbf{pt}_c))/T)}, \tag{1}$$

where $K$ is the number of categories, $\cos(\cdot, \cdot)$ denotes cosine similarity, and $T$ is a temperature parameter.

**Cross-domain open-vocabulary object detection.** We consider multiple labeled source datasets $\{\mathcal{D}^s = \{I_i^s, y_i^s\}_{i=1}^{N_s}\}_{s=1}^{N}$ from domains $\{\mathbb{D}^s\}_{s=1}^{N}$, and an unlabeled target dataset $\mathcal{D}^t = \{I_i^t\}_{i=1}^{N_t}$ from domain $\mathbb{D}^t$, where $N_s$ and $N_t$ denote the number of samples in source and target domains, respectively. For each image $I_i^s \in \mathcal{D}^s$, its annotation is $y_i^s = \{b_i^s, c_i^s\}$, with bounding-box coordinates $b_i^s$ and class labels $c_i^s \in \mathbb{C}^s$. Here, $\mathbb{C}^s$ denotes the label space of domain $\mathbb{D}^s$ containing $K_s$ predefined categories. We define the base label space as the union of all source-domain label sets, i.e., $\mathbb{C}^{\text{base}} = \bigcup_{s=1}^{N} \mathbb{C}^s$. The target label space is denoted by $\mathbb{C}^t$, which includes both base and novel categories: $\mathbb{C}^t = \mathbb{C}^{\text{base}} \cup \mathbb{C}^{\text{novel}}$, with $\mathbb{C}^{\text{novel}} \cap \mathbb{C}^{\text{base}} = \varnothing$. Here, $\mathbb{C}^{\text{novel}}$ represents categories not seen in any source domain, however, the category names can be utilized for domain adaptation. For simplicity, we denote the total number of categories across domains by $K$.

## 3.1 MULTI-SOURCE PROMPT LEARNING

We present Multi-Source Prompt Learning (MSPL) for MAP, which learns class- and domain-specific tokens using a standard domain-adaptive prompt structure that separates domain-invariant semantics from domain-specific patterns for better cross-domain generalization. Prompts are optimized with a class-specific loss $\mathcal{L}_{\text{class}}^s$ to align with category semantics and a domain-specific loss $\mathcal{L}_{\text{domain}}^s$ to capture domain variations. As shown in Section 4, these objectives are theoretically grounded and provably contribute to effective prompt learning.

**Source prompt design.** Following the domain-adaptive prompt formulation, each source-specific prompt is defined as:

$$\mathbf{pt}_c^s = \mathbb{V}^c \mathbb{V}^s [\mathbf{c}][\mathbf{dl}_s], \tag{2}$$

where $\mathbb{V}^c = [\mathbf{v}_1^c][\mathbf{v}_2^c]...[\mathbf{v}_{M_1}^c]$ contains $M_1$ learnable tokens and $\mathbf{c} = e(\text{``CLASS-}c\text{''})$ is the tokenized class label. The domain-specific component $\mathbb{V}^s = [\mathbf{v}_1^s][\mathbf{v}_2^s]...[\mathbf{v}_{M_2}^s]$ is combined with a tokenized domain label $\mathbf{dl}_s = e(\text{``Domain-}s\text{''})$. While $\mathbb{V}^c$ and $[\mathbf{c}]$ capture category semantics shared across domains, $\mathbb{V}^s$ and $[\mathbf{dl}_s]$ encode domain-specific variations independent of categories. This structure allows each prompt to jointly model general category knowledge and domain-specific patterns, providing a strong foundation for cross-domain open-vocabulary detection.

**Learning class-specific tokens.** To learn class-specific domain invariant tokens $\mathbb{V}^c$ and $[\mathbf{c}]$, we propose to optimize the standard cross-entropy loss over each source domain for an object region $\mathbf{r}^s$ of category $k$:

$$\mathcal{L}_{\text{class}}^s = \mathbb{E}_{\mathcal{D}^s}[-\log p_{\text{class}}^s(\hat{y} = k|\mathbf{r}^s)], \tag{3}$$

where $\mathbf{r}^s$ an object region in an image of the $s$-th source domain. The classification probability $p_{\text{class}}^s$ is obtained by aligning the visual region features $f(\mathbf{r}^s)$ and corresponding *class* prompt $g(\mathbf{pt}_k^s)$:

$$p_{\text{class}}^s(\hat{y} = k|\mathbf{r}^s) = \frac{\exp(\cos(f(\mathbf{r}^s), g(\mathbf{pt}_k^s))/T)}{\sum_{c=1}^{K} \exp(\cos(f(\mathbf{r}^s), g(\mathbf{pt}_c^s))/T)}, \tag{4}$$

By minimizing Equation (3) across all $N$ source domains, class-level semantic knowledge ($\mathbb{V}^c$ and $[\mathbf{c}]$) is distilled into each source prompt $\mathbf{pt}_c^s$, enabling accurate category-level discrimination within its domain and promoting domain-invariant understanding.

**Learning domain-specific tokens.** To learn domain-specific tokens $\mathbb{V}^s$ and $[\mathbf{dl}_s]$ for each source domain prompt, we minimize the following domain-specific loss function:

$$\mathcal{L}_{\text{domain}}^s = \mathbb{E}_{\mathcal{D}^s}[-\log p_{\text{domain}}^s(\hat{y} = k|\mathbf{r}^s)], \tag{5}$$

where the probability formulation $p_{\text{domain}}^s$ defined in Equation (6) is a softmax over all class-domain prompt combinations, promoting the *domain* and *class* alignment with the visual region feature:

$$p_{\text{domain}}^s(\hat{y} = k|\mathbf{r}^s) = \frac{\exp(\cos(f(\mathbf{r}^s), g(\mathbf{pt}_k^s))/T)}{\sum_{d=1}^{N} \sum_{c=1}^{K} \exp(\cos(f(\mathbf{r}^s), g(\mathbf{pt}_c^d))/T)}. \tag{6}$$

By minimizing Equation (5), we encourage the source region $\mathbf{r}^s$ to be close with domain-specific prompt $\mathbf{pt}_k^s$ corresponding to its true class label $k$ in the vision language space, while simultaneously suppressing similarity with prompts from other domains ($d \neq s$). This contrastive design in Equation (6) enforces discriminative associations between domain-specific prompts and their corresponding visual regions, facilitating the learning of fine-grained, domain-aware representations. A formal theoretical analysis is presented in Section 4, which rigorously justifies these objectives and establishes a principled, theoretically grounded framework for effective prompt learning.

## 3.2 UNSUPERVISED TARGET PROMPT LEARNING

We introduce Unsupervised Target Prompt Learning (UTPL) in MAP, which designs prompts for the target domain in COVD using a novel class mask to handle unseen categories. UTPL learns prompts without supervision via three objectives guiding class-specific, domain-specific, and novel-mask tokens. As detailed in Section 4, these objectives are theoretically justified, providing a principled framework for effective target prompt learning.

**Target domain prompt design**   To enable open-vocabulary detection in the target domain, we introduce a learnable novel mask $\mathbb{M} = [m_1][m_2], ...[m_{M_3}]$, which serves to specialize the prompt representation for novel classes. The target prompt is defined as:

$$\mathbf{pt}^t = \mathbb{V}^c \mathbb{V}^t \mathbb{M}[\mathbf{c}][\text{"}\mathbf{dl}_t\text{"}]. \tag{7}$$

Similar to the source prompt, $\mathbb{V}^c$ and $[\mathbf{c}]$ encodes class-specific context, while $\mathbb{V}^t$ and $[\text{"}\mathbf{dl}_t\text{"}]$ captures target-domain-specific context. The mask $\mathbb{M}$ is designed to enhance attention toward novel categories in the target domain by providing an explicit cue to the model, helping it distinguish between seen and unseen categories and thus improving open-vocabulary detection performance.

To learn the target prompt without supervision, instead of relying on potentially unreliable pseudo labels, we propose an unsupervised target prompt optimization objective that encourages consistency in the model's predictions across different augmentations of the same image.

**Discussion of target image augmentation.**   Various image augmentation techniques—such as random cropping, translation, rotation, and Gaussian noise—are commonly used in vision tasks. However, for COVD, it is crucial to preserve the semantic integrity of the image and minimize noise to maintain accurate predictions. In the multi-source domain adaptation setting, we aim to fully leverage pre-trained source models. To this end, we propose narrowing the domain gap by augmenting target images with source domain styles. Traditional methods like AdaIN Huang & Belongie (2017) can transfer the style from a source image to a target image. More recently, text-guided augmentation techniques Fahes et al. (2023); Kwon & Ye (2022); Suresh et al. (2024) have enabled style transfer based on textual descriptions, offering greater flexibility without requiring explicit source images—thus preserving source domain privacy. In our work, we adopt CLIPstyler Kwon & Ye (2022), a text-guided augmentation method, for target image style transfer. Formally, given an image $I \in \mathcal{D}^t$, we generate a stylized image $\mathcal{A}_s(I)$ using CLIPstyler with a style text prompt $T_s$. Additional augmentation details and visual examples are provided in the Appendix.

After obtaining stylized target augmentations, the learning of target prompt is formulated in an unsupervised manner with the following three parts:

**Learning class-specific tokens.**   To learn class-specific tokens for the unsupervised target domain, we propose minimizing the entropy of the averaged prediction probability distribution across different prompts and augmentations. The objective is formulated as:

$$\mathcal{L}_{\text{class}}^t = \mathbb{E}_{\mathcal{D}^t}[-\sum_{k=1}^{K} \tilde{p}_{\text{class}}(\hat{y} = k|\mathbf{r}^t) \log \tilde{p}(\hat{y} = k|\mathbf{r}^t)], \tag{8}$$

where $\tilde{p}_{\text{class}}(\hat{y} = k|\mathbf{r}^t) = \frac{1}{N+1} \sum_{s=0}^{N} p_{\text{class}}^t(\hat{y} = k|\mathcal{A}_s(\mathbf{r}^t)$ denotes the average prediction probability across $N$ augmented views and the original view. Each prediction is computed by aligning (stylized) visual region feature and its corresponding *class* prompt:

$$p_{\text{class}}^t(\hat{y} = k|\mathcal{A}_s(\mathbf{r}^t)) = \frac{\exp\left(\cos(f(\mathcal{A}_s(\mathbf{r}^t)), g(\mathbf{pt}_k^s))/T\right)}{\sum_{c=1}^{K} \exp\left(\cos(f(\mathcal{A}_s(\mathbf{r}^t)), g(\mathbf{pt}_c^s))/T\right)}, \tag{9}$$

where $\mathcal{A}_s(I)$ is an augmented view of $I$ with $s$ source's style, and $\mathcal{A}_0(\mathbf{r}^t)$ denotes the original view. Minimizing $\mathcal{L}_{\text{class}}^t$ encourages the model to produce consistent and confident (i.e., low-entropy) predictions across different domain-style augmentations of the same target image, each paired with its corresponding source-specific prompts. By enforcing agreement across these views, the model is incentivized to extract features that are stable under stylistic shifts. This effectively forces the class-specific tokens to capture semantic information that is invariant across diverse domain styles.

**Learning domain-specific tokens.**   To facilitate the learning of target domain-specific tokens, we minimize the following entropy loss over all prompt–region pairs:

$$\mathcal{L}_{\text{domain}}^t = \mathbb{E}_{\mathcal{D}^t}[-\sum_{k=1}^{K} \tilde{p}_{\text{domain}}(\hat{y} = k|\mathbf{r}^t) \log \tilde{p}_{\text{domain}}(\hat{y} = k|\mathbf{r}^t)], \tag{10}$$

where $\tilde{p}_{\text{domain}}(\hat{y} = k|\mathbf{r}^t) = \frac{1}{N+1} \sum_{s=0}^{N} p_{\text{domain}}^t(\hat{y} = k|\mathcal{A}_s(\mathbf{r}^t))$ and the prediction probability promotes the alignment of *domain* and *class* with the (stylized) visual region feature:

$$p_{\text{domain}}^t(\hat{y} = k|\mathcal{A}_s(\mathbf{r}^t)) = \frac{\exp\left(\cos(f(\mathcal{A}_s(\mathbf{r}^t)), g(\mathbf{pt}_k^s))/T\right)}{\sum_{d=0}^{N} \sum_{c=1}^{K} \exp\left(\cos(f(\mathcal{A}_d(\mathbf{r}^t)), g(\mathbf{pt}_c^d))/T\right)}. \tag{11}$$

Note that the denominator sum over all source and target domain, where $d = 0$ indicates the target domain. Similar to the formulation in Equation (6), the probability expression in Equation (11) pushes the model to select the correct class–domain pair under entropy minimization, while discouraging the alignment of mismatched *domain* and *class* prompts with the (stylized) visual region, thereby facilitating the unsupervised learning of the target domain-specific tokens.

**Learning novel class mask.** To learn the mask for novel classes, we propose encouraging the novel class prompts to diverge from the base class features by minimizing the orthogonality loss:

$$\mathcal{L}_{\text{novel}} = \sum_{c \in \mathbb{C}^{\text{base}}} \sum_{k \in \mathbb{C}^{\text{novel}}} (g(\mathbf{pt}_c)^\intercal g(\mathbf{pt}_k))^2 \qquad (12)$$

This minimizes dot product between base and novel prompt embeddings, making them directionally independent. Similarly, for all source and target prompts, to promote their distinctions, we also impose the mutual orthogonality on them:

$$\mathcal{L}_{\text{orth}} = \sum_{i \in \mathbb{P}^{\text{source}}} \sum_{i \in \mathbb{P}^{\text{target}}} \mathbb{1}_{\text{diff}}(i, j)(g(\mathbf{pt}^i)^\intercal g(\mathbf{pt}^j))^2, \qquad (13)$$

where we use $\mathbb{P}^{\text{source}}$ and $\mathbb{P}^{\text{target}}$ to denote the set of source and target prompts, respectively. $\mathbb{1}_{\text{diff}}(i, j) = 1$ when $i$ and $j$ are not the same class label (to avoid penalizing matched semantics). Minimizing $\mathcal{L}_{\text{orth}}$ encourages each prompt to be dissimilar to the rest, effectively promoting mutual orthogonality. This enforces that prompts encode distinct semantic information, thereby reducing redundancy and enhancing domain diversity.

### 3.3 TRAINING AND INFERENCE

During adaptation, the pre-trained visual backbones and the text encoder are kept frozen. In each iteration, only the learnable prompts are updated via backward gradients, which is performed by minimizing the overall learning objective $\mathcal{L}_{\text{total}}$, and the gradients can be back-propagated all the way through the text encoder $g(\cdot)$, making use of the rich knowledge encoded in the parameters to optimize the context. The the overall learning objective is defined as:

$$\mathcal{L}_{\text{total}} = (1 - \eta)(\mathcal{L}^{\text{src}} + \lambda \mathcal{L}^{\text{tgt}}) + \eta \mathcal{L}_{\text{orth}}, \qquad (14)$$

where $\mathcal{L}^{\text{src}} = \frac{1}{N} \sum_{s=1}^{N} \mathcal{L}_{\text{class}}^s + \mathcal{L}_{\text{domain}}^s$, and $\mathcal{L}^{\text{tgt}} = \mathcal{L}_{\text{class}}^t + \mathcal{L}_{\text{domain}}^t + \mathcal{L}_{\text{novel}}$. The source and target prompt loss $\mathcal{L}^{\text{src}}$ and $\mathcal{L}^{\text{tgt}}$ play equally important roles during cross-domain adaptation. The additional orthogonal regularization term $\mathcal{L}_{\text{orth}}$, balanced by a hyperparameter $\lambda$, brings mutual orthogonality among different domain prompts, enforcing distinction for each domain prompt. After training, the learned prompts are saved and used for inference.

## 4 THEORETICAL ANALYSIS

In this section, we provide a theoretical analysis of the proposed prompts through the lens of their associated loss functions. Inspired by Wang et al. (2024a), we analyze the behavior of the prompts from two perspectives: fidelity and distinction. Fidelity describes a prompt's ability to faithfully encode either domain-specific and class-specific information by maintaining strong alignment with the label semantics of its corresponding domain or class. That is, a prompt with high fidelity captures the underlying semantics of the class it represents or preserving the characteristics of the domain it originates from. Distinction, on the other hand, requires minimal overlap between prompts from different domains or classes, ensuring that each prompt captures unique domain or class-specific characteristics without redundancy. Prompts with high distinction are mutually dissimilar, which facilitates better disentanglement of semantic and domain-specific information and reduces ambiguity during alignment.

To formalize these concepts, we first introduce propositions for both fidelity (including both supervised and unsupervised formulations for class and domain prompts) and distinction (including both inter-class and inter-domain variants). Based on these formulations, we analyze our proposed prompts from these two perspectives.

**Definition 4.1** (**Fidelity**). A prompt $\mathbf{pt}$ is said to have high *fidelity* if it retains task-relevant semantic information that aligns closely with the target label $y$. Formally, fidelity can be measured by the mutual information $\mathrm{MI}(\mathbf{pt}, y)$ between the prompt and the label. A prompt with high fidelity satisfies a high $\mathrm{MI}(\mathbf{pt}, y)$, indicating strong semantic alignment with class-specific and domain-specific cues.

**Definition 4.2** (**Distinction**). A prompt $\mathbf{pt}^i$ is said to exhibit *distinction* if it captures information that is discriminative for its associated class and domain, while remaining minimally redundant with prompts from other classes or domains. Formally, distinction can be quantified by the (negative) mutual information with other prompts $\mathbf{pt}^j$ ($j \neq i$): $-\mathrm{MI}(\mathbf{pt}^i, \mathbf{pt}^j)$, where lower mutual information indicates higher distinction. Specifically, for class-specific prompts, this is referred to as *inter-class distinction*. For domain-specific prompts, this is referred to as *inter-domain distinction*.

**Definition 4.3** (**Unsupervised Fidelity**). A prompt $\mathbf{pt}$ is said to exhibit *unsupervised fidelity* if it induces confident and semantically meaningful predictions in the absence of ground-truth labels. Given an unlabeled input region $\mathbf{x} \in \mathcal{X}$, the predicted class distribution is denoted by $p(\hat{y}|\mathbf{pt}, \mathbf{x})$. Assuming a non-degenerate label distribution, unsupervised fidelity is measured by the negative Shannon entropy of the prediction distribution:

$$\mathrm{UnFI}(\mathbf{pt}) = -H(p(\hat{y}|\mathbf{pt}, \mathbf{x})), \tag{15}$$

where $H(p) = -\sum_k p_k \log p_k$. Lower entropy reflects higher confidence and alignment with semantic content, indicating greater fidelity.

Based on the definitions introduced in Definition 4.1 through Definition 4.3, we establish the following propositions to analyze the behavior of the proposed prompt tokens with respect to fidelity, unsupervised fidelity, and distinction, including both inter-class and inter-domain distinctions:

**Proposition 4.4** (**Source Class-Specific Prompt and $\mathcal{L}^s_{\mathbf{class}}$, $\mathcal{L}_{\mathbf{orth}}$**). *For source class-specific prompt, minimizing $\mathcal{L}^s_{class}$ achieves high fidelity, and low inter-domain distinction. By jointly minimizing the orthogonality loss $\mathcal{L}_{orth}$, explicit high inter-class distinction is enforced.*

**Proposition 4.5** (**Source Domain-Specific Prompt and $\mathcal{L}^s_{\mathbf{domain}}$, $\mathcal{L}_{\mathbf{orth}}$**). *For source domain-specific prompt, minimizing $\mathcal{L}^s_{domain}$ achieves high domain-specific fidelity and low inter-class distinction. By jointly minimizing the orthogonality loss $\mathcal{L}_{orth}$, explicit high inter-domain distinction is enforced.*

**Proposition 4.6** (**Target Class-specific Prompt and $\mathcal{L}^t_{\mathbf{class}}$, $\mathcal{L}_{\mathbf{orth}}$**). *For target class-specific prompt, minimizing $\mathcal{L}^t_{class}$ achieves high unsupervised fidelity and low inter-domain distinction. By jointly minimizing $\mathcal{L}_{orth}$, explicit high inter-class distinction is enforced.*

**Proposition 4.7** (**Target Domain-specific Prompt**). *For target domain-specific prompt, minimizing $\mathcal{L}^t_{domain}$ achieves high unsupervised fidelity and low inter-class distinction. By jointly minimizing $\mathcal{L}_{orth}$, explicit high inter-domain distinction is enforced.*

**Proposition 4.8** (**Novel Mask Prompt**). *For novel mask prompt, minimizing $\mathcal{L}_{novel}$ enforce high inter-class distinction among novel and base classes.*

Through Proposition 4.4 and Proposition 4.8, we establish a theoretical foundation for designing prompts that generalize across domains while preserving semantic separability among classes. Proofs are deferred to Appendix E.

## 5 EMPIRICAL EVALUATION

**Datasets.** We perform extensive experiments on datasets, including (1) the Art Image dataset with different artistic styles including Clipart1k, Comic2k, and Watercolor2k Inoue et al. (2018); (2) Diverse Weather Dataset (DWD) Wu & Deng (2022), and (3) Cityscapes Cordts et al. (2016), FoggyCityscapes Sakaridis et al. (2018), and KITTI Geiger et al. (2013a). Clipart1k contains 1000 clipart images across 20 classes, Watercolor2k and Comic2k contains 2000 watercolor/comic images across 6 classes. DWD covers five domains: Day Clear (DC), Night Clear (NC), Dusk Rainy (DR), Night Rainy (NR), and Day Foggy (DF). Each domain collects images from different weather and lighting conditions. During adaptation, we detect novel classes beyond the six classes, such as traffic light, and train, which do not appear in the source domains, as open-vocabulary evaluation. Cityscapes and FoggyCityscapes contains 7 classes, KITTI contains synthetic images of cars.

**Evaluation metrics.** Mean Average Precision (mAP) is used as the evaluation metric in all our experiments. We report mAP@0.5, which considers a prediction as a true positive if it matches the ground-truth label and has an intersection over union (IOU) score of more than 0.5 with the ground-truth bounding box. To evaluate performance across different class groups, we report APnovel and APbase, which correspond to the average precision computed over novel and base classes, respectively.

**Baselines.** We conduct our experiment based on four state-of-the-art open-vocabulary object detectors: RegionCLIP Zhong et al. (2022), OVMR Ma et al. (2024), YOLO-World Cheng et al. (2024) and GDINO Liu et al. (2024). Based on the object detectors, we implement different domain adaptive prompt learning methods including CoOP Zhou et al. (2022a), DAPL Ge et al. (2023), DAPro Li et al. (2024), and multi-source domain adaptive prompt learning methods including MPA Chen et al. (2023), SAP-CLIP Yang et al. (2024a), APNE-CLIP Yang et al. (2024b) and POND Wang et al. (2024a) as well as lower bound (LB) which adapts pre-trained open-vocabulary object detector to the target domain with handcraft prompt " A photo of a [CLASS]".

## 5.1 MAIN RESULTS

In Tab. 1, we provide the quantitative performance of our proposed framework MAP on four open-vocabulary object detectors Zhong et al. (2022); Ma et al. (2024); Cheng et al. (2024); Liu et al. (2024), and compared with the state-of-the-art domain adaptive prompt learning methods Zhou et al. (2022b); Ge et al. (2023); Li et al. (2024) and multi-source domain adaptive prompt learning methods Chen et al. (2023); Yang et al. (2024b;a); Wang et al. (2024a) on the art image. First, we observe that MAP significantly outperforms LB (i.e., the original object detector applied directly to the target domain without adaptation), as LB fails to address the substantial domain gap present in the target domain. Second, MSPL methods, including MAP, demonstrate a clear advantage over traditional DAPL methods that rely on a single source domain, highlighting the benefit of leveraging multiple source domains for more robust domain adaptation. Third, MAP exhibits a clear advantage over other MSPL methods, particularly on novel classes, achieving nearly a 5% improvement over the next-best MSPL approach, demonstrating the effectiveness of the novel mask prompt.

In Table 2, we report the mAP results across the four domains in the DWD benchmark. Due to space limitations, APbase and APnovel are provided in the appendix. MAP consistently outperforms both DAPL and MSPL baselines across all domains and backbone detectors. The detailed results in the appendix further demonstrate that the detection of novel classes is significantly improved, highlighting the effectiveness of the proposed novel mask prompt.

Table 1: Domain adaptation results (APs) on Clipart1k. For multi-source domain adaptation methods (highlighted with *), Watercolor2k and Comic2k are used jointly as source domains. For single-source domain adaptation methods, either Watercolor2k or Comic2k is used as the source domain, and only the best performance is reported. LB denotes Lower Bound, where the backbone detector is directly applied to the target domain without adaptation.

| Backbone | RegionCLIP Zhong et al. (2022) | | | OVMR Ma et al. (2024) | | | YOLO-World Cheng et al. (2024) | | | GDINO Liu et al. (2024) | | |
|---|---|---|---|---|---|---|---|---|---|---|---|---|
| Methods | AP | AP$_{base}$ | AP$_{novel}$ | AP | AP$_{base}$ | AP$_{novel}$ | AP | AP$_{base}$ | AP$_{novel}$ | AP | AP$_{base}$ | AP$_{novel}$ |
| LB | 41.14 | 42.85 | 39.03 | 42.86 | 43.33 | 40.57 | 41.96 | 42.87 | 39.83 | 43.06 | 43.94 | 40.79 |
| CoOP Zhou et al. (2022b) | 41.26 | 42.98 | 39.18 | 43.05 | 43.55 | 40.72 | 42.12 | 43.02 | 39.97 | 43.19 | 44.05 | 41.03 |
| DAPL Ge et al. (2023) | 41.57 | 43.25 | 39.47 | 43.36 | 43.86 | 41.05 | 42.54 | 43.41 | 40.42 | 43.48 | 44.49 | 41.48 |
| DAPro Li et al. (2024) | 42.42 | 44.13 | 40.36 | 44.29 | 44.75 | 41.93 | 43.47 | 44.33 | 41.35 | 44.38 | 45.40 | 42.37 |
| MPA* Chen et al. (2023) | 45.38 | 47.05 | 43.34 | 47.08 | 47.65 | 44.84 | 46.42 | 47.30 | 44.28 | 47.35 | 48.26 | 45.19 |
| SAP* Yang et al. (2024a) | 46.57 | 48.31 | 44.42 | 48.14 | 48.72 | 45.96 | 47.51 | 48.42 | 45.41 | 48.44 | 49.35 | 46.28 |
| APNE* Yang et al. (2024b) | 46.88 | 48.65 | 44.85 | 48.47 | 49.18 | 46.34 | 47.92 | 48.85 | 45.83 | 48.89 | 49.78 | 46.72 |
| POND* Wang et al. (2024a) | 49.54 | 51.36 | 47.45 | 51.12 | 51.89 | 49.02 | 50.58 | 51.49 | 48.49 | 51.50 | 52.47 | 49.45 |
| **MAP**\*(Ours) | **52.54** | **53.25** | **52.18** | **55.35** | **56.42** | **55.14** | **54.86** | **55.98** | **54.42** | **55.55** | **56.67** | **55.16** |

## 5.2 ABLATION STUDY

In this section, we present the ablative studies to verify the effectiveness of MAP, including different components of the prompt design (Table 3), and different architectures. Additional ablatives about the impact of hyperparameters are presented in the Appendix.

**Effectiveness of each loss functions.** In Table 3, we present an ablation study demonstrating that each loss component contributes to improved cross-domain open-vocabulary adaptation. Introducing a learnable class prompt via $\mathcal{L}_{class}^s$ improves performance by nearly 2% compared to fixed prompts, confirming the benefit of adapting category semantics. Incrementally adding more components yields steady improvements, with the combination of both class- and domain-level objectives for

Table 2: Domain adaptation results (mAP) on DWD. Following typical setting Wu & Deng (2022), *Day Clear* is used as source domain. The rest three are added as source domains for building a multi-source setting for each target domain for multi-source domain adaptation methods.

| Backbone | RegionCLIP Zhong et al. (2022) | | | | OVMR Ma et al. (2024) | | | | YOLO-World Cheng et al. (2024) | | | | GDINO Liu et al. (2024) | | | |
|---|---|---|---|---|---|---|---|---|---|---|---|---|---|---|---|---|
| Methods | NC | DR | NR | DF | NC | DR | NR | DF | NC | DR | NR | DF | NC | DR | NR | DF |
| LB | 40.32 | 29.54 | 26.02 | 33.23 | 41.42 | 30.38 | 26.56 | 33.75 | 41.54 | 30.75 | 26.78 | 34.53 | 42.18 | 33.22 | 29.25 | 37.03 |
| CoOP Zhou et al. (2022b) | 42.24 | 31.75 | 28.15 | 35.48 | 43.22 | 32.44 | 28.43 | 35.93 | 43.65 | 32.58 | 28.75 | 36.37 | 44.53 | 33.55 | 29.55 | 37.26 |
| DAPL Ge et al. (2023) | 42.65 | 31.84 | 28.47 | 35.65 | 43.38 | 32.69 | 28.87 | 36.23 | 43.98 | 32.87 | 29.13 | 36.84 | 44.96 | 33.97 | 29.98 | 38.05 |
| DAPro Li et al. (2024) | 42.43 | 31.86 | 28.17 | 35.53 | 42.81 | 32.62 | 28.66 | 35.98 | 43.84 | 33.46 | 28.81 | 37.27 | 44.54 | 34.18 | 29.55 | 38.07 |
| MPA* Chen et al. (2023) | 43.58 | 33.04 | 29.35 | 36.75 | 43.99 | 33.75 | 29.81 | 37.13 | 45.03 | 34.52 | 29.94 | 38.39 | 45.72 | 35.30 | 30.67 | 39.24 |
| SAP* Yang et al. (2024a) | 43.62 | 33.09 | 29.39 | 36.72 | 43.98 | 33.76 | 29.85 | 37.17 | 45.06 | 38.42 | 29.98 | 38.55 | 45.77 | 35.33 | 30.69 | 39.26 |
| APNE* Yang et al. (2024b) | 43.65 | 33.14 | 29.13 | 36.78 | 44.05 | 33.82 | 29.93 | 37.24 | 45.13 | 34.62 | 30.05 | 38.48 | 45.84 | 35.40 | 30.74 | 39.33 |
| POND* Wang et al. (2024a) | 43.63 | 33.12 | 29.10 | 36.76 | 44.03 | 33.80 | 29.90 | 37.21 | 45.12 | 34.60 | 30.01 | 38.45 | 45.80 | 35.37 | 30.71 | 39.29 |
| **MAP*(Ours)** | **44.84** | **34.31** | **30.56** | **38.06** | **45.42** | **34.85** | **30.92** | **38.47** | **46.22** | **35.94** | **31.45** | **39.93** | **47.13** | **37.05** | **32.27** | **40.97** |

source and target domains leading to an overall gain of about 10%. While using only class or only domain prompts improves performance, the effect is less pronounced. Incorporating the novel class loss $\mathcal{L}_{\text{novel}}$ further boosts novel class detection by an additional $+1.8\%$, validating the effectiveness of explicitly modeling unseen categories. Finally, adding the orthogonality regularization $\mathcal{L}_{\text{orth}}$ contributes another $+1.5\%$, showing that encouraging disentanglement between class and domain prompts further enhances generalization.

Table 3: Ablative analysis each loss function on Clipart dataset.

| $\mathcal{L}^s_{\text{class}}$ | $\mathcal{L}^s_{\text{domain}}$ | $\mathcal{L}^t_{\text{class}}$ | $\mathcal{L}^t_{\text{domain}}$ | $\mathcal{L}_{\text{novel}}$ | $\mathcal{L}_{\text{orth}}$ | AP | AP$_{\text{base}}$ | AP$_{\text{novel}}$ |
|---|---|---|---|---|---|---|---|---|
| ✗ | ✗ | ✗ | ✗ | ✗ | ✗ | 43.45 | 43.94 | 40.79 |
| ✓ | ✗ | ✗ | ✗ | ✗ | ✗ | 45.42 | 45.76 | 40.94 |
| ✓ | ✓ | ✗ | ✗ | ✗ | ✗ | 45.55 | 45.88 | 41.57 |
| ✓ | ✓ | ✓ | ✗ | ✗ | ✗ | 46.12 | 46.51 | 42.49 |
| ✓ | ✓ | ✓ | ✓ | ✗ | ✗ | 53.24 | 53.87 | 52.25 |
| ✓ | ✗ | ✓ | ✗ | ✗ | ✗ | 44.18 | 44.48 | 39.69 |
| ✗ | ✓ | ✗ | ✓ | ✗ | ✗ | 45.08 | 45.25 | 40.32 |
| ✓ | ✓ | ✓ | ✓ | ✓ | ✗ | 54.07 | 53.60 | 54.06 |
| ✓ | ✓ | ✓ | ✓ | ✓ | ✓ | 55.55 | 56.67 | 55.16 |

**The influence of $\lambda$.** We conduct the experiment on Day Foggy dataset, where $\lambda$ range from 0.25 to 1.50. We present the mAP with different $\lambda$ values in the appendix (Figure 7), which indicates that too small $\lambda$ such as 0.25 lead to a sub-optimal performance since the target data is not fully utilized. While setting $\lambda = 1.0$, the learning from both source and target is balanced and lead to the optimal performance. In Figure 2, we present some qualitative results of using some of the $\lambda$ values. A small $\lambda$ value such as 0.5 misclassified `bike` as `Motor`, and miss some smaller objects due to insufficient utilization of target information such as unlabeled novel class images. $\lambda = 0.75$ is able to detect `Bike` but still not able to detect smaller objects. Similarly, $\lambda = 1.25$ failed to detect smaller objects or separates multiple `Persons` since it puts too much focus on learning the target knowledge. While setting $\lambda = 1.00$, the learning reaches a balance and lead to optimal performance.

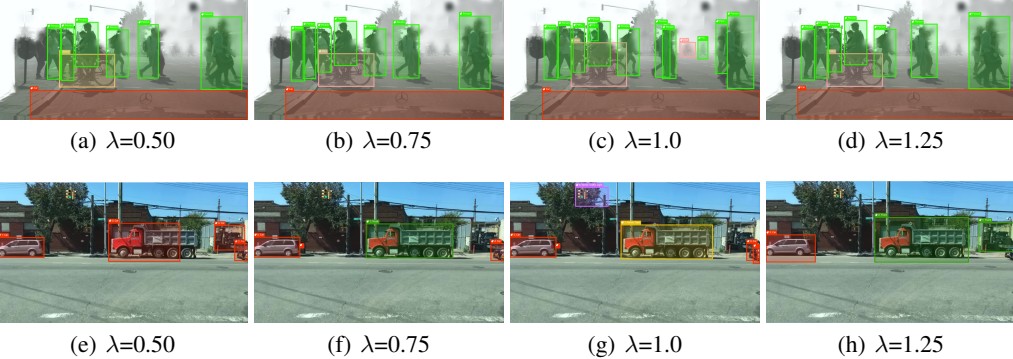

| (a) $\lambda$=0.50 | (b) $\lambda$=0.75 | (c) $\lambda$=1.0 | (d) $\lambda$=1.25 |
|---|---|---|---|
| (e) $\lambda$=0.50 | (f) $\lambda$=0.75 | (g) $\lambda$=1.0 | (h) $\lambda$=1.25 |

Figure 2: Detection results with different $\lambda$'s. The second row shows different $\lambda$'s influence to the detection of the novel class. (Traffic light in the illustrative figure.)

## 6 CONCLUSION

In this work, we tackle the unique challenges of cross-domain open-vocabulary object detection, where domain shift and category shift are intricately entangled. To address these challenges, we propose MAP, a parameter-efficient Multi-source prompt learning framework. MAP unifies Multi-Source Prompt Learning (MSPL) and Unsupervised Target Prompt Learning (UTPL) to effectively leverage diverse knowledge from multiple source domains and adapt to the unlabeled target domain via learnable prompts. We further provide a theoretical analysis of the proposed prompts in terms of their fidelity and distinction properties. Extensive experiments on cross-domain benchmarks demonstrate the effectiveness of our approach. In future work, we would work on developing techniques for the interpretability and explainability of the learned prompts for better model understanding.

## 7 REPRODUCIBILITY STATEMENT

We have taken multiple steps to ensure the reproducibility of our work. A detailed description of our proposed method and training objectives is provided in Section 3 of the main paper. Additional pseudo code of the proposed algorithm, detailed training steps, implementation details, hyperparameter settings, and dataset information are included in Appendix B. To further facilitate reproducibility, we provide an anonymous link to the source code and scripts for training and evaluation in Appendix G. All datasets used in our experiments are publicly available, and their references are properly provided.

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

# Supplementary Material

# Appendix

## Table of Contents

## A    TRAINING ALGORITHM AND NOTATIONS

The notations used in the main paper are shown in Table 4. And the learning process of MAP is presented in Algorithm 1.

Table 4: Summary of Major Notations

| Notation | Description |
|---|---|
| $s$ | $s$-th source domain |
| $t$ | Target domain |
| $N$ | Number of source domains |
| $K_s$ | Length of label space in domain $d$ |
| $K_t$ | Length of label space in target domain $t$ |
| $\mathbb{D}^s$ | $s$-th source domain |
| $\mathbb{D}^t$ | target domain $t$ |
| $\mathcal{D}^s$ | Dataset of $s$-th source domain $\mathbb{D}^s$ |
| $\mathcal{D}^t$ | Dataset of target domain $\mathbb{D}^t$ |
| $I_i^s$ | The $i$-th image of $s$-th source domain |
| $y_i^s$ | The $i$-th label of $s$-th source domain |
| $b_i^s$ | The bounding box of the $i$-th image of $s$-th source domain |
| $c_i^s$ | The class label of the $i$-th image of $s$-th source domain |
| $I_i^t$ | The $i$-th image of the target domain $t$ |
| $\mathbb{C}^s$ | Label space of $s$-th domain, $|\mathbb{C}^s| = K_s$ |
| $\mathbb{C}^t$ | Label space of target domain $t$ |
| $\mathbb{C}^{\text{base}}$ | Base classes |
| $\mathbb{C}^{\text{novel}}$ | Novel classes |
| $\mathbb{V}^c$ | The class-specific domain-invariant tokens for class $c$ |
| $\mathbb{V}^d$ | The domain-specific class-agnostic tokens for domain $s_d$ |
| [``domain$_s$''] | Handcrafted textual domain descriptions related to domain $s$ |
| $M_1$ | Length of the learnable tokens of $\mathbb{V}^c$ |
| $M_2$ | Length of the learnable tokens of $\mathbb{V}^d$ |
| $M_3$ | Length of the learnable tokens of $\mathbb{M}$ |
| $\mathbf{pt}_c^s$ | The prompt for class $c$ in source domain $s$ |
| $\mathbf{pt}_c^t$ | The prompt for class $c$ in target domain $t$ |
| $f(\cdot)$ | Visual encoder |
| $g(\cdot)$ | Text encoder |
| $N_r$ | Number of region boxed for image $I$ |
| $\mathbf{r}_I^j$ | $j$-th region of image $I$ |
| $\mathbf{f}_I^j$ | $j$-th region box embedding of image $I$ |
| $\cos(\cdot, \cdot)$ | cosine similarity |

## B    IMPLEMENTATION DETAILS

**Model Architecture.**    In this work, we tested four open-vocabulary object detectors. Each source model is initialized with a pre-trained open-vocabulary object detector. They are then fine-tuned using their own labeled source datasets, and adapted to the target domain via MAP. During pre-training, the text encoders are kept frozen, while the visual encoders are fine-tuned using the labeled source data. During adaptation, each fine-tuned visual encoder is used to process source images from its corresponding domain. The target visual encoder is initialized as a vanilla visual encoder.

**Implementation details.**    We adopt the open-vocabulary object detector backbone to encode the input images and their corresponding text encoder to encode the input text prompts. The length of learnable tokens $M_1$, $M_2$ and $M_3$ are fixed as 8. The hyperparameter $\lambda$ is set to 1. We use SGD optimizer with learning rate equals to 0.002. Following Ge et al. (2023), we randomly initialize each prompt with a zero-mean Gaussian distribution with a standard deviation of 0.02. The training of the prompts and the inference are conducted on one NVIDIA V100 GPU. The augmented images are generated with the pre-trained CLIPstyler model Kwon & Ye (2022).

---

**Algorithm 1** Training Procedure of MAP

---

1: **INPUT** Source datasets $\{\mathcal{D}^s\}_{s=1}^N$, target dataset $\mathcal{D}^t$, domain style text prompt $\{T_s\}_{d=1}^N$, and target domain style text prompt $T_t$
2: Initialize source and target prompts as "A photo of [CLS]"
3: Obtain source prompts and target prompt embedding using text encoder $g(\cdot)$: $\{\{\mathbf{pt}_c^s\}_{s=1}^N\}_{c=1}^K$, and $\{\mathbf{pt}_c^t\}_{c=1}^K$
4: **while** not converged **do**
5:   **for** $d \in \{1, ..., N\}$ **do**
6:     Sample a source batch $\mathcal{B}_s \sim \mathcal{D}^s$
7:     Compute $\mathcal{L}_{\text{domain}}^s$ and $\mathcal{L}_{\text{class}}^s$ using Equation (5) and Equation (3), respectively
8:   **end for**
9:   Sample a target batch $\mathcal{B}_t \sim \mathcal{D}^t$
10:   **for** each image $I_t$ in $\mathcal{B}_t$ **do**
11:     **for** each style description in $\{T_s\}_{d=1}^N$ **do**
12:       Augment $I_t$ with $T_s$ and $T_t$ to obtain $\mathcal{A}_s(I_t)$
13:     **end for**
14:   **end for**
15:   Compute $\mathcal{L}_{\text{domain}}^t$ and $\mathcal{L}_{\text{class}}^t$ using Equation (10) and Equation (8), respectively
16:   Compute $\mathcal{L}_{\text{novel}}$ using Equation (12)
17:   Compute $\mathcal{L}_{\text{orth}}$ using Equation (13)
18:   Compute overall loss and update $\mathbf{pt}^s$ and $\mathbf{pt}^t$
19: **end while**

---

**Text-guide Synthesis.** Text-guide Synthesis aim to generated images with different styles given an input base image and a text description. CLIPstyler Kwon & Ye (2022) design a modulation of the style of content images only with a single text condition using the pre-trained text-image embedding model of CLIP, and propose a patch-wise text-image matching loss with multiview augmentations for realistic texture transfer. PODA Fahes et al. (2023) propose Prompt-driven Instance Normalization (PIN) to learn style statistics based on CLIP feature alignment which are later used for image augmentation. PromptStyler Cho et al. (2023) simulates various distribution shifts in the joint space by synthesizing diverse styles via prompts without using any images to deal with source-free domain generalization. In this work, we use CLIPstyler Kwon & Ye (2022) to generate augmentations of unlabeled target imagesto learn target prompts. Formally, CLIPStyler aims to transfer the semantic style of target text $t_{sty}$ to the content image $I_c$ through the pre-trained text-image embedding model CLIP without a style image $I_s$ to as a reference. Given the semantic text style of the style target and the input content $t_{sty}$ and $t_{src}$, CLIPStyler transforms the content image $I_c$ to stylized image $I_{cs}$. In our setting, we treat the unlabeled target image as the content image $I_c$, the target domain semantic description as $t_{src}$, and the source domain semantic description as $t_{sty}$. With multiple source domains available, we augment the target image with multiple styles. The corresponding domain descriptions are shown in Tab. 5. The augmentation results are shown in Fig. 3.

Table 5: Domain descriptions of DWD

| Domain Name | Domain Description |
|---|---|
| *Day Clear* | A driving in a clear day photo |
| *Day Foggy* | A driving in a foggy day photo |
| *Night rainy* | A driving in a rainy night photo |
| *Night clear* | A driving in a clear night photo |
| *Dusk rainy* | A driving in a rainy dusk photo |

## C  DETAILED ILLUSTRATION OF LOSS FUNCTIONS

In this section, we provide a detailed illustration of the learning objectives used in MAP through accompanying figures.

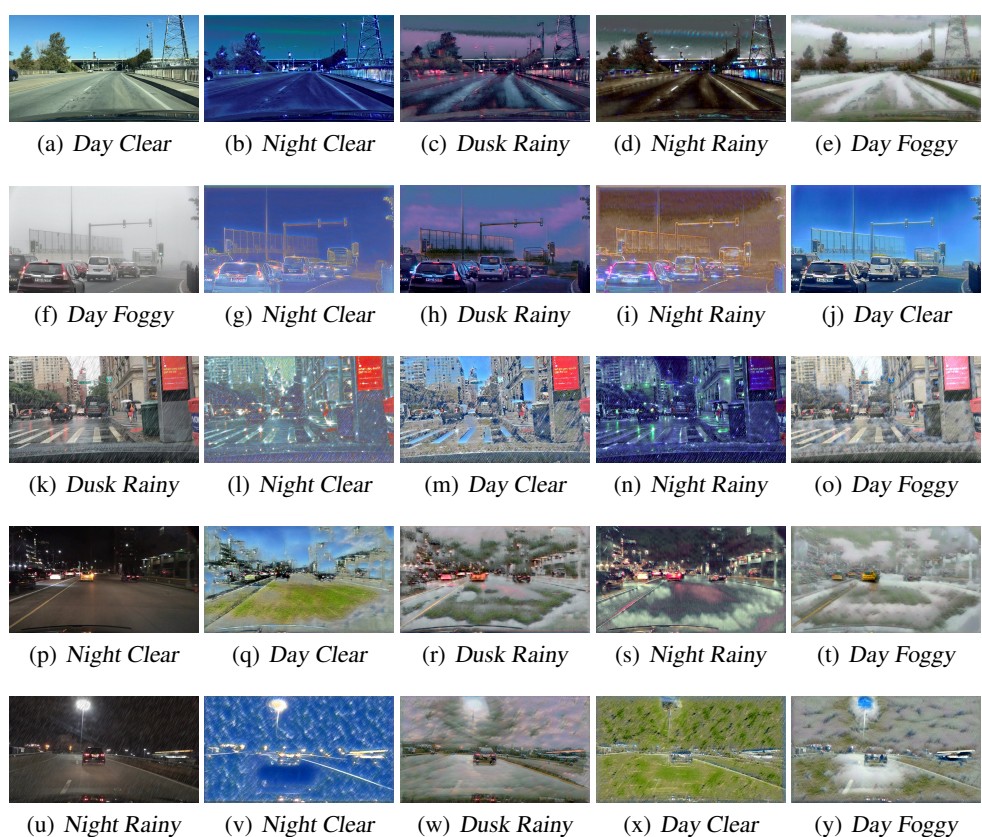

| (a) *Day Clear* | (b) *Night Clear* | (c) *Dusk Rainy* | (d) *Night Rainy* | (e) *Day Foggy* |
| (f) *Day Foggy* | (g) *Night Clear* | (h) *Dusk Rainy* | (i) *Night Rainy* | (j) *Day Clear* |
| (k) *Dusk Rainy* | (l) *Night Clear* | (m) *Day Clear* | (n) *Night Rainy* | (o) *Day Foggy* |
| (p) *Night Clear* | (q) *Day Clear* | (r) *Dusk Rainy* | (s) *Night Rainy* | (t) *Day Foggy* |
| (u) *Night Rainy* | (v) *Night Clear* | (w) *Dusk Rainy* | (x) *Day Clear* | (y) *Day Foggy* |

Figure 3: Stylized images of different styles with image sampled from different domains of DWD dataset wit descriptions shown in Tab. 5.

### C.1 ILLUSTRATION OF MSPL LEARNING OBJECTIVES

In Figure 4, we present figurative illustration for $\mathcal{L}_{\text{class}}^s$ and $\mathcal{L}_{\text{domain}}^s$. To differentiate $\mathcal{L}_{\text{class}}^s$ and $\mathcal{L}_{\text{domain}}^s$, we use "Src Prompt 1 Emb" and "Src Prompt N Emb" to specify different source prompt embeddings outputed by the text encoder. As shown in Figure 4, both $\mathcal{L}_{\text{class}}^s$ and $\mathcal{L}_{\text{domain}}^s$Both losses take a visual region embedding (omitted in the figure for clarity, please refer to Figure 1) and the corresponding source prompt as input. The key difference is that $\mathcal{L}_{\text{domain}}^s$ incorporates prompt embeddings from other source domains in addition to its own, promoting the learning of domain-aware representations.

### C.2 ILLUSTRATION OF UTPL LEARNING OBJECTIVES

In Figure 5, we illustrate the loss functions of learning target class-specific and domain specific tokens. Both are learned in an unsupervised manner with augmented views and source prompts involved. In Figure 6, we demonstrate the loss functions of the novel class mask and the prompt orthogonality.

## D ADDITIONAL EXPERIMENTAL RESULTS

In this section, we present additional $\text{AP}_{\text{base}}$ and $\text{AP}_{\text{novel}}$ results of the DWD benchmark in Table 6 and Table 7. We present ablative for hyper-parameters including $\lambda$, $M_1$, $M_2$ and $M_3$ in Appendix D.3. The visualization of prompt embeddings are shown in Appendix D.5. Additional experiments on datasets including Cityscapes, FoggyCityscapes, and KITTI are presented in Appendix D.7.

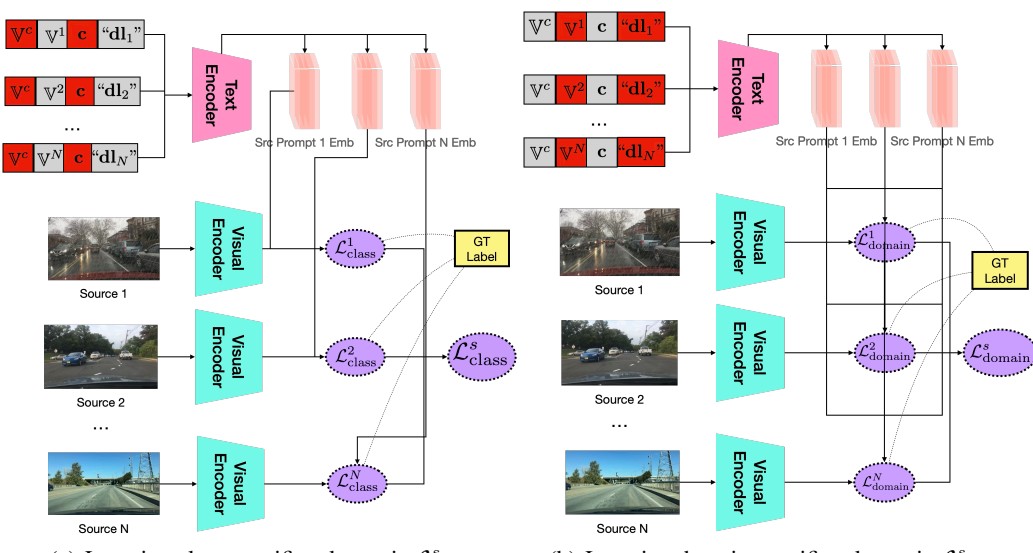

(a) Learning class-specific tokens via $\mathcal{L}_{class}^s$.

(b) Learning domain-specific tokens via $\mathcal{L}_{domain}^s$.

Figure 4: Illustration of source prompt learning objectives: $\mathcal{L}_{class}^s$ and $\mathcal{L}_{domain}^s$.

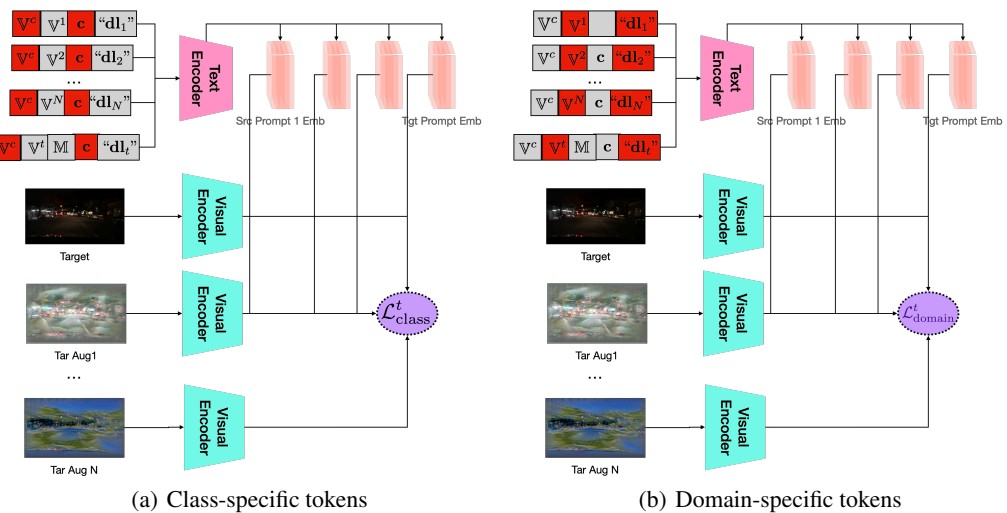

(a) Class-specific tokens

(b) Domain-specific tokens

Figure 5: Illustration of learning target domain prompts.

## D.1 $AP_{BASE}$ AND $AP_{NOVEL}$ RESULTS OF THE DWD BENCHMARK

In Table 6 and Table 7, we present the $AP_{base}$ and $AP_{novel}$. We observe that MAP has ranked the top across different object detector backbones and different domains. Similarly, MSPL methods performs better than SSPL methods due to the utilization of multiple source models. But our model outperform other MSPL methods since other methods only focus on addressing domain shift while neglecting the category shift.

## D.2 ABLATION STUDY

**Target model initialization.** For simplicity, our previous experiments initialized the target model using a vanilla visual encoder. In this ablation study, we investigate the effect of initializing the target model with a source model fine-tuned on one of the source domains. This approach can fully leverage prior knowledge from a particular source domain and may accelerate adaptation, especially when

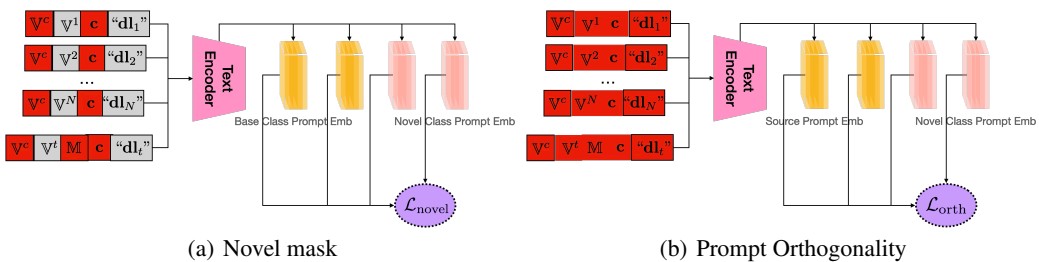

(a) Novel mask      (b) Prompt Orthogonality

Figure 6: Illustration of class and domain orthogonality loss.

Table 6: Domain adaptation results ($AP_{base}$) on DWD with the same setting as Table 2.

| Backbone | RegionCLIP Zhong et al. (2022) | | | | OVMR Ma et al. (2024) | | | | YOLO-World Cheng et al. (2024) | | | | GDINO Liu et al. (2024) | | | |
|---|---|---|---|---|---|---|---|---|---|---|---|---|---|---|---|---|
| Methods | NC | DR | NR | DF | NC | DR | NR | DF | NC | DR | NR | DF | NC | DR | NR | DF |
| LB | 44.56 | 33.25 | 29.45 | 36.79 | 44.98 | 34.56 | 30.11 | 36.98 | 45.15 | 34.68 | 30.09 | 37.88 | 45.92 | 37.05 | 32.82 | 40.82 |
| CoOP Zhou et al. (2022b) | 45.78 | 34.66 | 31.59 | 38.67 | 46.54 | 35.92 | 31.74 | 38.55 | 46.83 | 35.62 | 31.58 | 39.75 | 46.84 | 37.12 | 33.04 | 40.95 |
| DAPL Ge et al. (2023) | 45.57 | 34.94 | 31.82 | 37.74 | 47.05 | 37.01 | 31.95 | 39.52 | 47.05 | 36.02 | 33.05 | 40.03 | 48.06 | 37.05 | 33.08 | 41.97 |
| DAPro Li et al. (2024) | 45.56 | 34.89 | 31.22 | 38.62 | 45.78 | 35.69 | 31.75 | 39.02 | 46.95 | 36.58 | 31.95 | 40.32 | 47.68 | 37.22 | 33.69 | 41.15 |
| MPA* Chen et al. (2023) | 46.89 | 36.45 | 32.58 | 39.98 | 47.15 | 36.87 | 32.96 | 40.25 | 48.24 | 37.85 | 33.04 | 41.52 | 48.88 | 38.65 | 33.84 | 42.41 |
| SAP* Yang et al. (2024a) | 46.78 | 36.12 | 32.45 | 39.84 | 47.01 | 36.84 | 32.89 | 40.20 | 48.15 | 37.54 | 33.09 | 41.74 | 48.95 | 38.51 | 33.79 | 42.42 |
| APNE* Yang et al. (2024b) | 47.14 | 37.25 | 33.08 | 40.82 | 48.08 | 37.85 | 33.95 | 41.25 | 49.12 | 38.70 | 34.11 | 42.52 | 49.85 | 39.45 | 34.90 | 43.38 |
| POND* Wang et al. (2024a) | 47.42 | 37.32 | 33.12 | 40.85 | 48.11 | 37.90 | 33.98 | 41.41 | 49.18 | 38.65 | 34.15 | 42.63 | 49.90 | 39.46 | 34.89 | 43.40 |
| **MAP*(Ours)** | **47.45** | **37.38** | **33.18** | **40.92** | **48.16** | **37.98** | **34.09** | **41.49** | **49.24** | **38.77** | **34.22** | **42.78** | **50.02** | **39.58** | **34.95** | **43.51** |

the target domain shares visual or semantic similarity with that source. However, such initialization may also introduce domain bias, potentially leading to overfitting to the source characteristics and suboptimal performance when the source-target domain gap is large.

To study this effect, we perform experiments on the Clipart1k target domain, using three initialization strategies: a vanilla object detector, a source model fine-tuned on Comic2k, and a source model fine-tuned on Watercolor2k. Results across four backbone detectors (RegionCLIP, OVMR, YOLO-World, and GDINO) are shown in Table 8. We observe that initializing with a Comic2k or Watercolor2k model generally improves AP slightly over the vanilla initialization, especially when the backbone benefits from domain-specific knowledge without suffering from strong domain bias. This highlights the trade-off between leveraging source-specific priors and maintaining generalization.

**Heterogeneous source model architecture.** For simplicity, prior experiments used the same object detector architecture across all source models. However, our framework naturally extends to heterogeneous source backbones, as source models remain frozen during adaptation. In this ablation study, we explore this flexibility by employing a combination of four distinct backbone detectors: RegionCLIP, OVMR, YOLO-World, and GDINO, each pre-trained on a different source domain—Night Rainy, Dusk Rainy, Day Clear, and Night Clear, respectively. We adapt these models to the target domain Day Foggy. The resulting performance achieves an AP of 39.40%, with APbase at 40.89% and APnovel at 38.77%. While this is slightly lower than using a single strong backbone such as YOLO-World or GDINO alone—due to the relatively weaker performance of RegionCLIP and OVMR—it demonstrates that heterogeneous backbone integration is feasible and effective. Overall, this experiment confirms that MAP is compatible with multi-source models of varying architectures, enabling greater flexibility in practical deployment.

Table 7: Domain adaptation results ($AP_{novel}$) on DWD with the same setting as Table 2.

| Backbone | RegionCLIP Zhong et al. (2022) | | | | OVMR Ma et al. (2024) | | | | YOLO-World Cheng et al. (2024) | | | | GDINO Liu et al. (2024) | | | |
|---|---|---|---|---|---|---|---|---|---|---|---|---|---|---|---|---|
| Methods | NC | DR | NR | DF | NC | DR | NR | DF | NC | DR | NR | DF | NC | DR | NR | DF |
| LB | 36.95 | 25.98 | 22.96 | 29.93 | 38.05 | 27.15 | 23.35 | 30.52 | 38.39 | 27.50 | 23.62 | 30.98 | 38.95 | 30.18 | 26.19 | 34.02 |
| CoOP Zhou et al. (2022b) | 38.79 | 27.89 | 24.29 | 31.79 | 39.54 | 28.82 | 24.56 | 32.12 | 39.85 | 28.78 | 24.91 | 32.59 | 40.84 | 29.78 | 26.89 | 35.82 |
| DAPL Ge et al. (2023) | 38.85 | 27.94 | 24.35 | 31.86 | 39.62 | 28.88 | 24.61 | 32.18 | 39.95 | 28.88 | 24.98 | 32.68 | 40.95 | 29.87 | 26.99 | 35.94 |
| DAPro Li et al. (2024) | 38.86 | 27.96 | 24.37 | 31.89 | 39.65 | 28.94 | 24.68 | 32.22 | 39.98 | 28.91 | 25.04 | 32.72 | 40.79 | 30.36 | 27.05 | 35.96 |
| MPA* Chen et al. (2023) | 40.44 | 29.95 | 26.18 | 33.55 | 40.89 | 30.54 | 26.65 | 34.02 | 41.94 | 31.39 | 26.58 | 34.89 | 42.61 | 33.21 | 27.58 | 36.19 |
| SAP* Yang et al. (2024a) | 40.68 | 30.14 | 26.37 | 33.81 | 41.15 | 30.76 | 26.88 | 34.26 | 42.19 | 31.67 | 26.74 | 35.15 | 42.82 | 33.44 | 27.80 | 36.41 |
| APNE* Yang et al. (2024b) | 40.84 | 30.32 | 26.55 | 34.03 | 41.35 | 30.98 | 27.02 | 34.47 | 42.41 | 31.89 | 26.95 | 35.38 | 43.04 | 33.65 | 28.02 | 36.65 |
| POND* Wang et al. (2024a) | 41.25 | 30.78 | 26.96 | 34.51 | 41.78 | 31.42 | 27.48 | 34.90 | 42.85 | 32.31 | 27.38 | 35.80 | 43.49 | 34.09 | 28.44 | 37.10 |
| **MAP*(Ours)** | **42.75** | **32.19** | **28.42** | **35.83** | **43.22** | **32.66** | **28.87** | **36.31** | **44.15** | **33.79** | **29.36** | **37.90** | **45.09** | **35.02** | **30.50** | **38.24** |

Table 8: Domain adaptation results (APs) on Clipart1k using different pre-trained models to initialize the target model.

| Backbone | RegionCLIP Zhong et al. (2022) | | | OVMR Ma et al. (2024) | | | YOLO-World Cheng et al. (2024) | | | GDINO Liu et al. (2024) | | |
|---|---|---|---|---|---|---|---|---|---|---|---|---|
| Methods | AP | $AP_{base}$ | $AP_{novel}$ | AP | $AP_{base}$ | $AP_{novel}$ | AP | $AP_{base}$ | $AP_{novel}$ | AP | $AP_{base}$ | $AP_{novel}$ |
| Vanilla | 52.54 | 53.25 | 52.18 | 55.35 | 56.42 | 55.14 | 54.86 | **55.98** | 54.42 | 55.55 | 56.67 | 55.16 |
| Comic2k | 52.69 | 53.45 | **52.36** | **55.99** | **56.66** | **55.70** | **54.89** | 55.67 | **54.55** | 56.28 | 56.71 | 56.09 |
| Watercolor2k | **52.90** | **54.80** | 52.08 | 55.79 | 56.06 | 55.67 | 54.79 | 55.69 | 54.41 | **56.40** | **56.83** | **56.22** |

**Computational Complexity Analysis.** Table 9 presents a comparison of the inference time and memory usage of our method against existing baseline approaches. For a fair comparison, we evaluate MAP alongside other MSPL methods. As shown, our method achieves the fastest inference speed at 74.1 FPS and the smallest parameter size of 37M, demonstrating both computational efficiency and lower memory footprint.

Table 9: Time and Memory Complexity

| | MPA* Chen et al. (2023) | SAP* Yang et al. (2024a) | APNE* Yang et al. (2024b) | POND* Wang et al. (2024a) | **MAP***(Ours) |
|---|---|---|---|---|---|
| Inference Time (FPS) | 63.2 | 69.7 | 60.5 | 52.0 | 74.1 |
| Parameter Size (M) | 45 | 40 | 46 | 48 | 37 |

**Impact of different augmentation methods.** We evaluate several augmentation strategies, including CLIPstyler Kwon & Ye (2022), CycleGAN Zhu et al. (2017), UNIT Liu et al. (2017), and UNSB Kim et al. (2023). As shown in Table 10, CycleGAN achieves the best overall AP with a slight gain of $+0.23\%$ over our baseline, suggesting that richer pixel-level stylization can marginally enhance domain realism. However, the improvement is limited, and text-guided augmentation remains competitive while being more lightweight, controllable, and free from style image requirements.

Table 10: Performance comparison of different methods.

| Metric | CLIPstyler | CycleGAN | UNIT | UNSB |
|---|---|---|---|---|
| AP | 55.55 | 55.78 | 54.89 | 55.42 |
| $AP_{base}$ | 56.67 | 57.23 | 56.45 | 56.79 |
| $AP_{novel}$ | 55.16 | 55.01 | 54.88 | 54.95 |

**Integrating LLMs in OVOD.** In this ablation, we examine the impact of leveraging more powerful pre-trained large language models (LLMs). Using YOLO-World as a baseline, we replace the original CLIP text encoder with the stronger LLaVA-OneVision-0.5b-ov Liu et al. (2023). Since the detector and the language model were pre-trained independently, we follow YOLO-World's pre-training strategy and re-train the detector for 5 epochs to align its visual features with the new text encoder. As shown in Table 11, this substitution yields consistent gains in detection accuracy, demonstrating that stronger LLM/MLLM backbones can significantly enhance cross-domain open-vocabulary detection. This result highlights the potential of integrating state-of-the-art multimodal language models into OVD frameworks to further push performance boundaries.

D.3 IMPACT OF HYPERPARAMETERS

In this section, we perform the experiments of different choices of the hyperparameters $\lambda$, $M_1$, $M_2$, $M_3$, and $\eta$ to explore their impact.

**The influence of $M_1$, $M_2$, $M_3$** We conduct experiments on the *Day Foggy* dataset using G-DINO as the model backbone, varying $M_1$ and $M_2$ from 6 to 12. Figure 7(a) presents a heatmap of mAP scores for different values of $M_1$ and $M_2$. The results indicate that values that are too small or too large result in sub-optimal performance, likely due to underfitting or overfitting. The best performance is achieved when both $M_1$ and $M_2$ are set to 8. To evaluate the influence of $M_3$, we fix $M_1$ and $M_2$ at 8 and vary $M_3$ from 6 to 12. The results show that the optimal performance is achieved when $M_3 = 9$. However, variations in $M_3$ do not significantly affect performance on the Day Foggy domain when $M_1$ and $M_2$ are fixed.

Table 11: Comparison between different vision-language models on cross-domain open-vocabulary detection.

| Method | AP | $AP_{base}$ | $AP_{novel}$ |
|---|---|---|---|
| YOLO-World with CLIP (LB) | 41.96 | 42.87 | 39.83 |
| YOLO-World with LLaVA | 42.88 | 43.85 | 40.70 |
| MAP with CLIP | 54.86 | 55.98 | 54.42 |
| MAP with LLaVA (Ours) | 55.19 | 56.45 | 54.68 |

Table 12: Impact of the hyperparameter $M_3$

| Length | 6 | 7 | 8 | 9 | 10 | 11 | 12 |
|---|---|---|---|---|---|---|---|
| AP | 40.41 | 40.55 | 40.85 | **40.97** | 40.95 | 40.82 | 40.69 |
| $AP_{base}$ | 43.04 | 43.19 | 43.37 | **43.51** | 43.48 | 43.30 | 43.17 |
| $AP_{novel}$ | 37.92 | 38.05 | 38.13 | **38.24** | 38.22 | 38.14 | 38.04 |

**The influence of $\lambda$** We conduct the experiment on *Day Foggy* dataset, where $\lambda$ range from 0.25 to 1.50. We present the mAP with different $\lambda$ values in Figure 7(b), which indicates that too small $\lambda$ such as 0.25 lead to a sub-optimal performance since the target data is not fully utilized. While setting $\lambda = 1.0$, the learning from both source and target is balanced and lead to the optimal performance.

**The influence of $\eta$.** We evaluate the effect of $\eta$ on the Clipart dataset with values ranging from 0.001 to 0.30, and report the results in Table 13. The best performance is obtained at $\eta = 0.10$, indicating that a small but non-zero regularization from the orthogonality term enhances performance. However, when $\eta$ becomes too large, the orthogonality term dominates training and significantly degrades both base and novel class accuracy.

### D.4 ADDITIONAL ABLATIONS

**Impact of different components.** We conduct three extra ablations to verify the effectiveness of text-guided style augmentation, the entropy minimization objective and the domain description respectively by (1) Removing text-guided style augmentation. We replace the text-guided source-style augmentation with generic augmentations (random cropping, translation, rotation, Gaussian noise). Because these augmentations do not preserve source-domain style, the domain gap between the source model and the augmented views increases, leading to a noticeable performance drop compared to using text-guided style augmentation. (2) Removing entropy minimization and consistency. We remove the target-domain losses includes $\mathcal{L}^t_{class}$, and $\mathcal{L}^t_{domain}$, and train UTPL only with $\mathcal{L}_{novel}$ $\mathcal{L}_{orth}$. Without entropy minimization and consistency, the target prompt cannot learn meaningful class-specific or domain-specific tokens. As a result, the remaining regularization terms are insufficient to guide the target prompt, and performance degrades significantly. (3) Replacing domain-specific

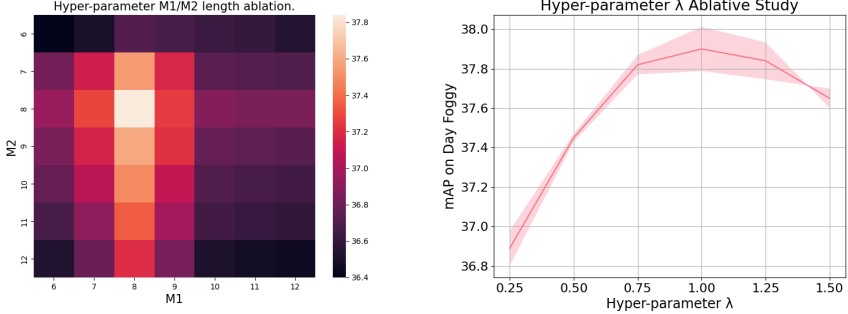

(a) Heatmap mAP on *Day Foggy* in terms of different lengths of the learnable tokens of $\mathbb{V}^c$ $(M_1)$ and $\mathbb{V}^d$ $(M_2)$.

(b) Multi-source test mAP on *Day Foggy* with $\lambda$ ablations.

Figure 7: Impact of Hyperparameters.

Table 13: Impact of the hyperparameter $\eta$.

| Metric | 0.001 | 0.01 | 0.10 | 0.30 |
|---|---|---|---|---|
| AP | 55.47 | 55.50 | **55.55** | 52.13 |
| $AP_{base}$ | 56.60 | 56.63 | **56.67** | 52.98 |
| $AP_{novel}$ | 55.05 | 55.07 | **55.16** | 51.69 |

descriptions with generic ones. Previously we use domain-specific descriptions such as "comic" (Comic2k) or "watercolor" (Watercolor2k). We replace them with generic descriptions: "art." These generic descriptions encode little domain information, so the target prompt is less domain-aware, which reduces adaptation performance.

| **Method Variant** | AP | $AP_{base}$ | $AP_{novel}$ |
|---|---|---|---|
| remove text guided style augmentation | 46.52 | 48.24 | 44.35 |
| remove entropy minimization and consistency | 41.88 | 42.63 | 39.59 |
| generic domain descriptions | 50.92 | 49.77 | 48.65 |

Table 14: Ablation study results.

**Comparison with unsupervised prompt learning methods.** To provide a more comprehensive evaluation, we have included comparisons with recent few-shot and unsupervised prompt tuning methods that are closely related to our setting Shu et al. (2022); He et al. (2023); Vuong et al. (2025). As shown in Table 15, these approaches represent strong and up-to-date baselines in unsupervised prompt tuning. Their performance is generally in line with other single-source prompt learning approaches. In contrast, our method explicitly leverages multiple heterogeneous source domains and integrates them through the meta-learned aggregation mechanism, which leads to substantially better performance across base and novel categories.

| **Method** | AP | $AP_{base}$ | $AP_{novel}$ |
|---|---|---|---|
| TPT Shu et al. (2022) | 46.37 | 47.45 | 45.52 |
| UPT He et al. (2023) | 44.33 | 44.95 | 43.61 |
| M-CRPL Vuong et al. (2025) | 40.28 | 41.35 | 38.84 |
| **MAP** | **55.55** | **56.67** | **55.16** |

Table 15: Comparison with recent unsupervised prompt tuning baselines.

**Performance of MAP on classification or segmentation tasks.** To further support the generality of the framework, we additionally conduct experiments on classification (DomainNet Leventidis et al. (2021), Table 16) and semantic segmentation (ACDC Sakaridis et al. (2021), Table 17), confirming that the same MAP formulation transfers well to other tasks without architectural changes. For both tasks, we follow the open-vocabulary protocol by removing a subset of classes from the source domain to ensure no label overlap with the target domain.

**Additional experiments for single-source methods with combined source domains.** For fairness comparison, we conducted additional experiments where all single-source methods are trained on the combined source domains, treating them as a single source. The results are presented in Table 18. These results show that incorporating information from multiple source domains indeed boosts the performance of SSDA methods. This further reinforces the importance of properly integrating heterogeneous sources, something that our proposed meta-learned aggregation framework is explicitly designed to address.

**Diffusion-based augmentation.** Our choice of CLIPStyler employs lightweight, controllable text-driven style augmentation. Here we consider diffusion-based style transfer methods. We conducted additional experiments using diffusion-based augmentation Islam et al. (2024); Niemeijer et al. (2024); Yang et al. (2023); Brooks et al. (2023). The results are summarized in Table 19. The results

| Domain | AP | AP$_{base}$ | AP$_{novel}$ |
|--------|------|------|------|
| Clipart | 92.88 | 94.45 | 91.52 |
| Infograph | 90.67 | 91.23 | 90.12 |
| Painting | 88.45 | 89.65 | 88.01 |
| Quickdraw | 89.98 | 88.79 | 90.24 |
| Real | 87.65 | 88.42 | 86.98 |
| Sketch | 91.23 | 91.88 | 90.47 |

Table 16: Classification Accuracy across DomainNet domains.

| Domain | AP | AP$_{base}$ | AP$_{novel}$ |
|--------|------|------|------|
| Foggy | 33.7 | 34.8 | 30.5 |
| Nighttime | 33.1 | 33.9 | 30.4 |
| Rainy | 31.5 | 32.4 | 28.2 |
| Snowy | 31.2 | 32.2 | 28.1 |

Table 17: Segmentation Performance across ACDC domains.

show that diffusion-based augmentations produce higher-quality stylized images and generally lead to improved adaptation performance, further confirming that style-based augmentation effectively reduces the domain gap. This also demonstrates that our framework is flexible and can directly benefit from more advanced generative models as they continue to improve.

### D.5 VISUALIZATION OF PROMPT EMBEDDINGS

In this ablation, we visualized the different components of the learned prompts. We first randomly select 75 sample prompts that summarize each pairing input image. By reducing their dimensionality into a 2-d plane with T-SNE, we visualize each textual prompt embedding from every domain along with the domain invariant and domain specific tokens generated in our proposed method, as shown in Figs 8, 9 and 10. We choose to visualize the embedding of the class `car` to show the domain shift effect and our method's capability to decompose such effect into a respective domain invariant token and a domain specific embedding token for each input image prompt. In Fig. 8, we first plot the image embeddings from *Day Clear* (i.e., source) and *Day Foggy* (i.e., target) with red and blue dots. The domain-invariant tokens are denoted with orange stars. We also plot the template embedding with handcrafted prompts such as "a photo of car in the road, in the street", etc and denote them as magenta pentagon. From Fig. 8 red circle grouped dots, we find that the model discovered domain invariant embedding resides between the source and target domains, and locate very near to the template. Similar finding can be seen in Fig. 9 with *Night Clear* and *Night Rainy* domains. All above findings confirm that our method could summarize effective domain invariant embeddings that grasp the major objects of the image. In Fig. 10, we plot the domain-specific tokens with different colors of stars: blue, green, red and purple. We also plot the handcrafted template in orange pentagon. From Fig. 10, we can also see that the domain-specific tokens reside near to each domain's shifted embedding while be far away from the template, for all four data domains. We have highlighted two groups with blue and red circles in Fig. 10 for better illustration.

### D.6 VISUALIZATION OF CLASS EMBEDDINGS

In Figure 11, we present the t-SNE visualization of base and novel classes on the Clipart dataset. Aside from a few outliers, the base and novel classes are well separated, indicating that the novel mask loss encourages the novel classes to explore a distinct feature space.

| Method | AP | AP$_{base}$ | AP$_{novel}$ |
|--------|------|--------|---------|
| LB | 46.14 | 47.08 | 43.85 |
| CoOP | 45.24 | 46.08 | 43.21 |
| DAPL | 46.54 | 47.58 | 44.62 |
| DAPro | 47.85 | 48.88 | 45.49 |

Table 18: Single-source methods trained on the combined source domains.

| Method | AP | AP_base | AP_novel |
|--------|------|---------|----------|
| Diffusemix [2] | 53.54 | 54.28 | 53.08 |
| DIDEX [3] | 48.54 | 48.99 | 47.62 |
| ZeCon [4] | 49.54 | 49.99 | 48.62 |
| InstructPix2Pix [5] | 53.78 | 55.18 | 53.70 |
| PODA [6] | 50.62 | 50.87 | 49.49 |

Table 19: Comparison of diffusion-based and style-transfer augmentations.

## D.7 ADDITIONAL DATASETS

We create a new multi-source dataset by simply utilizing **Cityscapes**[1] Cordts et al. (2016), **Foggy-Cityscapes**[2] Sakaridis et al. (2018), and **KITTI**[3] Geiger et al. (2013b) for further evaluation. Cityscapes consist of 2,975 training images and 500 testing images, have a total of 8 categories captured under normal weather. Foggy-Cityscapes applies images of Cityscapes to simulate foggy as well as inherits the annotations of Cityscapes. KITTI contains 7,481 urban images which are different from Cityscapes. The Cityscapes and KITTI are used as source domains, while FoggyCityscapes is the target domain. To augment the target images with the source domain styles, we use "A photo of driving in the city in a foggy daytime" as source description and "A photo of driving in the city during daytime" and "A photo of driving in the urban city" as target descriptions. For open-vocabulary object detection, we reserve `Rider`, `Bike`, `Motor` as novel classes. The performance is presented in Tab. 20. It's evident that our method excels in the single-source domain adaptation methods, attaining state-of-the-art results. Furthermore, when compared to other multi-source domain adaptive techniques, the advantage of our approach is evident across novel and base classes.

## D.8 ADDITIONAL METRIC

In this section, we provide AP AP@[.5:.95] for the GDINO backbone on Clipart1k. As shown below, the performance gain of our method is preserved under the stricter COCO-style metric.

---

[1] `https://github.com/tiancity-NJU/da-faster-rcnn-PyTorch`
[2] `https://www.cityscapes-dataset.com/downloads/`
[3] `http://www.cvlibs.net/datasets/kitti/`

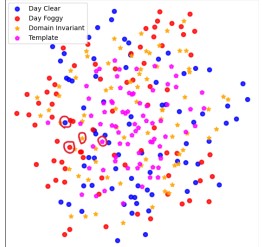

Figure 8: Day Clear versus Day Foggy Textual Embedding and Domain Invariant Tokens.

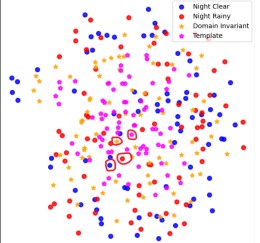

Figure 9: Night Clear versus Night Rainy Textual Embedding and Domain Invariant Tokens.

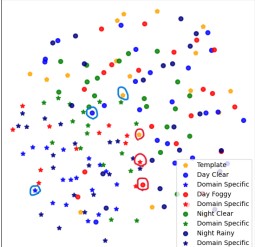

Figure 10: All domains textual embedding and their respective Domain Specific Tokens.

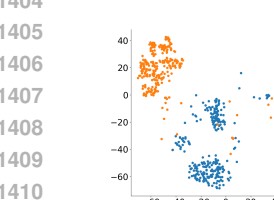

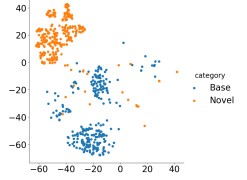

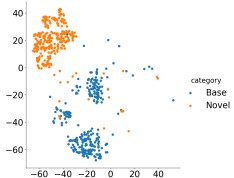

Figure 11: t-SNE of base and novel class in Clipart.

Figure 12: t-SNE of base and novel class in DWD Day Foggy.

Figure 13: t-SNE of base and novel class in Foggycityscapes.

Table 20: Domain adaptation results (APs) on Foggycityscapes. For multi-source domain adaptation methods (highlighted with *), Cityscapes and KITTI are used jointly as source domains. For single-source domain adaptation methods, either Cityscapes or KITTI is used as the source domain, and only the best performance is reported. LB denotes Lower Bound, where the backbone detector is directly applied to the target domain without adaptation.

| Backbone | RegionCLIP Zhong et al. (2022) | | | OVMR Ma et al. (2024) | | | YOLO-World Cheng et al. (2024) | | | GDINO Liu et al. (2024) | | |
|---|---|---|---|---|---|---|---|---|---|---|---|---|
| Methods | AP | AP$_{base}$ | AP$_{novel}$ | AP | AP$_{base}$ | AP$_{novel}$ | AP | AP$_{base}$ | AP$_{novel}$ | AP | AP$_{base}$ | AP$_{novel}$ |
| LB | 40.15 | 41.87 | 38.06 | 41.87 | 42.34 | 39.59 | 40.95 | 41.89 | 38.86 | 42.07 | 42.95 | 39.80 |
| CoOP Zhou et al. (2022b) | 40.25 | 41.97 | 38.18 | 42.05 | 42.57 | 39.74 | 41.11 | 42.03 | 38.96 | 42.20 | 43.07 | 40.05 |
| DAPL Ge et al. (2023) | 40.59 | 42.26 | 38.49 | 42.38 | 42.86 | 40.06 | 41.56 | 42.40 | 39.44 | 42.49 | 43.49 | 40.49 |
| DAPro Li et al. (2024) | 41.44 | 42.15 | 39.39 | 43.34 | 43.78 | 40.97 | 42.50 | 43.37 | 40.39 | 43.42 | 44.42 | 41.40 |
| MPA* Chen et al. (2023) | 44.15 | 45.14 | 42.33 | 46.15 | 46.64 | 43.79 | 45.52 | 46.35 | 43.32 | 46.36 | 47.29 | 44.23 |
| SAP* Yang et al. (2024a) | 45.26 | 47.05 | 43.24 | 47.02 | 47.58 | 44.75 | 46.32 | 47.28 | 44.33 | 47.28 | 48.29 | 45.24 |
| APNE* Yang et al. (2024b) | 45.78 | 47.60 | 43.76 | 47.38 | 48.12 | 45.30 | 46.86 | 47.80 | 44.78 | 47.85 | 48.72 | 45.66 |
| POND* Wang et al. (2024a) | 48.50 | 50.31 | 46.42 | 50.08 | 50.82 | 48.78 | 49.55 | 50.44 | 47.45 | 50.48 | 51.44 | 48.41 |
| **MAP***(Ours) | **52.02** | **52.76** | **51.69** | **54.87** | **55.89** | **54.62** | **54.15** | **55.26** | **53.74** | **54.88** | **55.98** | **54.39** |

# E  THEORETICAL JUSTIFICATION

## E.1  PROOF OF DEFINITION 4.3.

*Proof.* In the unsupervised setting, where ground-truth labels $y$ are unavailable, we cannot directly evaluate the mutual information $\text{MI}(\mathbf{pt}; y)$. Instead, we approximate *fidelity* by quantifying the information shared between the input region $\mathbf{x}$ and the predicted label $\hat{y}$, i.e., $\text{MI}(\mathbf{x}; \hat{y})$, where $\hat{y}$ is the model's output given prompt $\mathbf{pt}$. We argue that under the assumption of an approximately constant $H(\hat{y})$, low entropy of the prediction distribution $p(\hat{y}|\mathbf{x}, \mathbf{pt})$ implies high mutual information between the input $\mathbf{x}$ and the predicted label $\hat{y}$, thus serving as a proxy for high fidelity.

Formally, the mutual information between $\mathbf{x}$ and $\hat{y}$ is given by:

$$\text{MI}(x; \hat{y}) = H(\hat{y}) - H(\hat{y}|\mathbf{x}), \qquad (16)$$

where $H(\hat{y})$ is the marginal entropy over predictions across the dataset, and $H(\hat{y}|\mathbf{x})$ is the conditional entropy of predictions for a given input $\mathbf{x}$.

Minimizing the entropy of the prediction distribution:

$$\mathcal{L}_{\text{entropy}} = -\sum_k p(\hat{y} = k|\mathbf{x}, \mathbf{pt}) \log p(\hat{y} = k|\mathbf{x}, \mathbf{pt}) \qquad (17)$$

effectively reduces $H(\hat{y}|x)$. If $H(\hat{y})$ is kept high or remains approximately constant (e.g., via diverse predictions across the dataset), then the mutual information $\text{MI}(\mathbf{x}; \hat{y})$ increases:

$$\text{Low } H(\hat{y}|\mathbf{x}) \Rightarrow \text{High MI}(\mathbf{x}; \hat{y}). \qquad (18)$$

Hence, minimizing entropy encourages confident and consistent predictions, which implies that the prompt encodes meaningful and class-relevant information. This serves as an effective surrogate for *high fidelity* in the absence of label supervision:

$$\boxed{\text{Low entropy} \Rightarrow \text{High mutual information } I(\mathbf{x}; \hat{y}) \Rightarrow \text{High unsupervised fidelity}}. \qquad (19)$$

$\square$

| Method | AP | AP$_{\text{base}}$ | AP$_{\text{novel}}$ |
|--------|-----|------|------|
| LB | 35.47 | 36.52 | 33.48 |
| CoOP | 35.67 | 36.78 | 33.77 |
| DAPL | 35.96 | 37.21 | 34.18 |
| DAPro | 36.84 | 38.12 | 35.05 |
| MPA | 39.52 | 40.68 | 37.65 |
| SAP | 40.62 | 41.79 | 38.73 |
| APNE | 42.85 | 43.54 | 41.29 |
| POND | 46.17 | 46.95 | 45.08 |
| MAP | **50.29** | **50.88** | **48.79** |

Table 21: Comparison of methods on AP, $AP_{\text{base}}$, and $AP_{\text{novel}}$.

### E.2 PROOF OF PROPOSITION 4.4

*Proof.* An effective class-specific prompt should possess high fidelity, meaning it preserves label-relevant information, and should exhibit low distinction with respect to domains, i.e., remain invariant to domain-specific characteristics. The class-specific loss $\mathcal{L}_{\text{class}}^s$ in Equation (3) encourages the class-specific prompt to maximize the likelihood of correct class predictions across all regions and source domains. From the perspective of fidelity, minimizing $\mathcal{L}_{\text{class}}^s$ increases the mutual information $\text{MI}(\mathbf{pt}_k, y_k)$ between the class-specific prompt $\mathbf{pt}_k$ and the label $y_k$, as the prompt is explicitly optimized to recover semantic label information. Specifically,

$$\text{MI}(\mathbf{pt}_k, y_k) = H(y_k) - H(y_k|\mathbf{pt}_k) \tag{20}$$

where $H(y_k)$ is the entropy and $H(y_k|\mathbf{pt}_k)$ is the entropy of $H(y_k)$ conditioned on $\mathbf{pt}_k$. Since $H(y_k)$ is constant, maximizing the mutual information $\text{MI}(\mathbf{pt}_k, y_k)$ is equivalent to minimizing $H(y_k|\mathbf{pt}_k)$:

$$\text{Min} H(\hat{y}|\mathbf{x}) \Rightarrow \text{Max MI}(\mathbf{x}; \hat{y}). \tag{21}$$

At the same time, because this loss is aggregated over multiple domains without incorporating domain labels, it discourages the prompt from encoding domain-specific cues, thereby implicitly enforcing low distinction with respect to domain. Moreover, class-specific prompts should also exhibit inter-class distinction since class-specific prompts should be disentangled across classes to capture diverse semantic attributes without redundancy. And this is ensured by minimizing $\mathcal{L}_{\text{orth}}$, which promotes mutual orthogonality explicitly. □

### E.3 PROOF OF PROPOSITION 4.5

*Proof.* An effective domain-specific prompt should faithfully encode the unique characteristics of each source domain (high fidelity) while remaining distinguishable from prompts of other domains (high inter-domain distinction), and invariant to classes (low inter-class distinction). This ensures that the prompt captures domain-specific cues that aid in learning domain-aware visual representations, which is particularly useful for mitigating domain shifts in cross-domain detection. The domain-specific loss function $\mathcal{L}_{\text{domain}}^s$ in Equation (5) explicitly enforces these principles. First, the fidelity is promoted by minimizing the negative log-likelihood in Equation (5), where the visual regions $\mathbf{r}_{ij}^s$ are aligned with their respective domain-specific prompts $\mathbf{pt}_k^s$. By ensuring that the visual features closely align with the correct domain-specific prompt, we are maximizing the mutual information between the prompt and its corresponding label $k$. This encourages the prompt to preserve domain-relevant information, reinforcing the fidelity of the representation within its source domain. Second, distinction is addressed by the contrastive nature of Equation (6), which defines the alignment over the set of all possible class-domain prompt pairs. This normalization ensures that the domain-specific prompts from different domains are not highly similar, thus encouraging minimal overlap. By minimizing this overlap, the domain-specific prompts for each source domain are distinguished from each other, resulting in high inter-domain distinction. This allows the model to better separate the learned domain-specific representations and maintain robust performance across domains. The prompt orthogonality loss $\mathcal{L}_{\text{orth}}$ in Equation (13) explicitly enforces high distinction among prompts. Minimizing $\mathcal{L}_{\text{orth}}$ reduces mutual information $\text{MI}(\mathbf{pt}^i, \mathbf{pt}^j)$ between different prompts, ensuring that each prompt captures unique and domain-specific information. With the softmax over different classes, low inter-class distinction is enforced. □

### E.4 PROOF OF PROPOSITION 4.6

*Proof.* In the context of unsupervised target domain adaptation, learning effective class-specific prompts without access to labeled target data is critical. Similar to the source class-specific case, the target class-specific prompt is expected to exhibit high fidelity and low inter-domain distinction. First, the entropy minimization loss $\mathcal{L}_{\text{class}}^t$ in Equation (8) promotes high fidelity by encouraging prediction consistency across augmented views, while $\mathcal{L}_{\text{orth}}$ explicitly enforces orthogonality among different class prompts to reduce redundancy and enhance distinction. From an information-theoretic perspective, low entropy implies confident predictions, and class diversity assuming a roughly constant $H(\hat{y})$, this corresponds to high mutual information between the learned prompt and the predicted label. According to Definition 4.3, the class-specific prompt presents high unsupervised fidelity. Thus, the learned prompt is more likely to encode class-discriminative and semantically meaningful features. Second, by averaging predictions across various domain-stylized augmentations and prompt variants, the model implicitly encourages low distinction between the target class-specific prompt and prompts from source domains. This regularization effect reduces the influence of domain-specific variations, pushing the prompt toward a domain-invariant, semantics-focused representation. □

### E.5 PROOF OF PROPOSITION 4.7

*Proof.* Similar to source domain-specific prompts, the target domain-specific prompt should demonstrate both high fidelity and high inter-domain distinctiveness. Building on the analysis of target class-specific prompts, high unsupervised fidelity for the target domain-specific prompt can be achieved by optimizing $\mathcal{L}_{\text{domain}}^t$. Meanwhile, high inter-domain distinctiveness can be explicitly encouraged through $\mathcal{L}_{\text{orth}}$, as is done for the source domain-specific prompts. □

### E.6 PROOF OF PROPOSITION 4.8

*Proof.* Minimizing $\mathcal{L}_{\text{novel}}$ enhances the distinction between the novel class prompts and the base class space. By promoting orthogonality between base and novel classes in the learned feature space, it enables novel prompts to specialize in capturing previously unseen concepts. □

### E.7 GENERALIZATION GUARANTEE

In this section, we explore the generalization error of open-vocabulary domain adaptation under multi-source setting. Given $N$ source domains, weighted by $\boldsymbol{\alpha} = (\alpha_1, \alpha_2, ..., \alpha_N)$, $\sum_{i=1}^{N} \alpha_i = 1$, we examine the convex combination of training error $\hat{err}_{\boldsymbol{\alpha}}(h)$ from each source domain following the setting in Ben-David et al. (2010), where $h \sim \mathcal{H}$ is a hypothesis in the hypothesis space $\mathcal{H}$. Denoting the empirical error in the target domain as $\hat{err}_t(h)$, the generalization error of multi-source open-vocabulary domain adaptation is bounded by the following Theorem E.1.

**Theorem E.1.** *Given $N$ labeled source domains $S_1, ..., S_N$ and an unlabeled target domain $T$, let $\hat{err}_{\boldsymbol{\alpha}}(h)$ be the empirical $\boldsymbol{\alpha}$-weighted error of a hypothesis $h$, $\pi_{novel}^t$ be the class prior probability for the novel classes in the target domain, for any $\delta \in (0, 1)$, with probability at lease $1 - \delta$,*

$$\frac{\hat{err}_t(h)}{1 - \pi_{novel}^t} \leq \hat{err}_{\boldsymbol{\alpha}}(h) + \sum_{i=1}^{N} \alpha_i(2\lambda_i + d_{\mathcal{H}\Delta\mathcal{H}}(D_i, D_T)) + \sum_{i=1}^{N} \alpha_i \left( \frac{\hat{err}_{novel}^t(h)}{1 - \pi_{novel}^t} - \hat{err}_{novel}^{s_i}(h) \right)$$

(22)

*where $\lambda_i = \min_{h \in \mathcal{H}}\{err_t(h) + err_i(h)\}$. $D_i$, $D_T$ are the domain distributions for source domain $S_i$ and target domain $T$, respectively. $\hat{err}_{novel}^{s_i}(h)$ and $\hat{err}_{novel}^t(h)$ are the empirical risk of samples belong to the novel classes.*

Equation (22) contains three parts, convex combination of source error, domain discrepancy, and open-vocabulary difference. Detailed proof of Theorem E.1 is provided in as follows:

*Proof.* **Step 1.** Given a symmetric loss $\ell$, with $h \in \mathcal{H}$, the expected risks of the convex combination source and target is defined as:

$$err_{\boldsymbol{\alpha}}(h) = \sum_{i=1}^{N} \alpha_i \mathbb{E}_{(\mathbf{x},\mathbf{y}) \sim D_i} \ell(h(\mathbf{x}), \mathbf{y}).$$

(23)

$$err_t(h) = \mathbb{E}_{(\mathbf{x},\mathbf{y})\sim D_t}\ell(h(\mathbf{x}),\mathbf{y}) \tag{24}$$

The partial risk of known target classes is

$$err_t^*(h) = \frac{1}{1-\pi_{\mathrm{novel}}^t}\int_{\mathcal{X}\times\mathcal{Y}^s}\ell(h(\mathbf{x}),\mathbf{y})dP_{X^tY^t}(\mathbf{x},\mathbf{y}) \tag{25}$$

where $\mathcal{X}$ is the feature space, $\mathcal{Y}^s$ is the source label space, we suppose all the source domain share the same label space $\mathcal{Y}^s$ for simplification. $P_{X^tY^t}$ is the joint distribution of the target domain.

The partial risk for the unknown target classes is

$$err_t^{\mathrm{novel}}(h) = \int_{\mathcal{X}}\ell(h(\mathbf{x}),\mathbf{y}_{\mathrm{novel}})dP_{X^t|\mathbf{y}_{\mathrm{novel}}}(\mathbf{x}) \tag{26}$$

And we have

$$err_t(h) = \pi_{\mathrm{novel}}^t err_t^{\mathrm{novel}}(h) + (1-\pi_{\mathrm{novel}}^t)err_t^*(h) \tag{27}$$

Then

$$\frac{err_t(h)}{1-\pi_{\mathrm{novel}}^t} - err_{\boldsymbol{\alpha}}(h) = err_t^*(h) - err_{\boldsymbol{\alpha}}(h) + \frac{\pi_{\mathrm{novel}}^t}{1-\pi_{\mathrm{novel}}^t}err_t^{\mathrm{novel}}(h) \tag{28}$$

**Step 2.** Since we assume all the source domain share the same label space $\mathcal{Y}^s$, we have

$$\sum_{i=1}^{N}\alpha_i\int_{\mathcal{X}\times\mathcal{Y}^{s_i}}\ell(h(\mathbf{x}),\mathbf{y})dP_{X^sY^s}(\mathbf{x},\mathbf{y}) = \int_{\mathcal{X}\times\mathcal{Y}^s}\ell(h(\mathbf{x}),\mathbf{y})dP_{X^sY^s}(\mathbf{x},\mathbf{y}) \tag{29}$$

$$err_t^*(h) - err_{\boldsymbol{\alpha}}(h) = \int_{\mathcal{X}\times\mathcal{Y}^t}\ell(h(\mathbf{x}),\mathbf{y})dP_{X^tY^t|\mathcal{Y}^s}(\mathbf{x},\mathbf{y}) - \sum_{i=1}^{N}\alpha_i\int_{\mathcal{X}\times\mathcal{Y}^s}\ell(h(\mathbf{x}),\mathbf{y})dP_{X^{s_i}Y^s}(\mathbf{x},\mathbf{y})$$

$$\leq err_t^*(\tilde{h}) + \int_{\mathcal{X}\times\mathcal{Y}^t}\ell(\tilde{h}(\mathbf{x}),\mathbf{y})dP_{X^tY^t|\mathcal{Y}^s}(\mathbf{x},\mathbf{y})$$

$$+ err_{\boldsymbol{\alpha}}(\tilde{h}) - \sum_{i=1}^{N}\alpha_i\int_{\mathcal{X}\times\mathcal{Y}^s}\ell(\tilde{h}(\mathbf{x}),\mathbf{y})dP_{X^{s_i}Y^s}(\mathbf{x},\mathbf{y}) \tag{30}$$

where $\tilde{h}$ is any hypothesis in $\mathcal{H}$. According to Fang et al. (2020), we have

$$\int_{\mathcal{X}\times\mathcal{Y}^t}\ell(h(\mathbf{x}),\tilde{h}(\mathbf{x}))dP_{X^tY^t|\mathcal{Y}^s}(\mathbf{x},\mathbf{y}) = \int_{\mathcal{X}}\ell(h(\mathbf{x}),\tilde{h}(\mathbf{x}))dP_{X^t|\mathcal{Y}^s}(\mathbf{x},\mathbf{y}) \tag{31}$$

And for $i \in \{1,...,N\}$

$$\int_{\mathcal{X}\times\mathcal{Y}^s}\ell(h(\mathbf{x}),\tilde{h}(\mathbf{x}))dP_{X^{s_i}Y^s}(\mathbf{x}) = \int_{\mathcal{X}}\ell(h(\mathbf{x}),\tilde{h}(\mathbf{x}))dP_{X^{s_i}}(\mathbf{x}) \tag{32}$$

Based on above equations, we have

$$err_t^*(h) - err_{\boldsymbol{\alpha}}(h) \leq err_t^*(\tilde{h}) - err_{\boldsymbol{\alpha}}(\tilde{h}) + \sum_{i=1}^{N}\alpha_i|\int_{\mathcal{X}}\ell(h(\mathbf{x}),\tilde{h}(\mathbf{x}))dP_{X^t|\mathcal{Y}^s}(\mathbf{x},\mathbf{y}) - \int_{\mathcal{X}}\ell(h(\mathbf{x}),\tilde{h}(\mathbf{x}))dP_{X^{s_i}}(\mathbf{x})|$$

$$\leq err_t^*(\tilde{h}) - err_{\boldsymbol{\alpha}}(\tilde{h}) + \sum_{i=1}^{N}\alpha_i d_{\mathcal{H}}(P_{X^t|\mathcal{Y}^s}, P_{X^{s_i}}) \tag{33}$$

Hence, based on the definition of $\lambda_i$, and denoting $D_i = P_{X^{s_i}}$, $D_T = P_{X^t|\mathcal{Y}^s}$, we have,

$$err_t^*(h) - err_{\boldsymbol{\alpha}}(h) \leq \sum_{i=1}^{N}\alpha_i(2\lambda_i + d_{\mathcal{H}\Delta\mathcal{H}}(D_i, D_T)) \tag{34}$$

**Step 3.** In this step, we prove that

$$\frac{\pi_{\text{novel}}^t}{1 - \pi_{\text{novel}}^t} err_{\text{novel}}^t(h) \le d_{\mathcal{H}\Delta\mathcal{H}}(D_i, D_T) + \frac{R_{\text{novel}}^t(h)}{1 - \pi_{\text{novel}}^t} - R_{\text{novel}}^t(h) \tag{35}$$

This step is similar the proof in Fang et al. (2020). Please refer to the detailed in Fang et al. (2020).

By combining the results from Step 2 & 3, we have

$$\frac{\hat{err}_t(h)}{1 - \pi_{\text{novel}}^t} \le \hat{err}_{\boldsymbol{\alpha}}(h) + \sum_{i=1}^{N} \alpha_i(2\lambda_i + d_{\mathcal{H}\Delta\mathcal{H}}(D_i, D_T))$$

$$+ \sum_{i=1}^{N} \alpha_i \left( \frac{\hat{err}_{\text{novel}}^t(h)}{1 - \pi_{\text{novel}}^t} - \hat{err}_{\text{novel}}^{s_i}(h) \right) \tag{36}$$

$\square$

## F    POTENTIAL SOCIETAL IMPACT AND LIMITATIONS

Open-vocabulary domain adaptation with multiple sources has the potential to significantly expand its application in diverse settings with various constraints, such as heterogeneous and dynamic domains with scarce labels. The transformative potential of open-vocabulary models lies in their ability to generalize from limited data by leveraging pre-trained knowledge, reducing the necessity for large, domain-specific datasets. This approach allows models to remain relevant and effective in dynamic environments without extensive retraining. By minimizing the need for comprehensive data collection and labeling, open-vocabulary domain adaptation also lowers the costs associated with model development and maintenance. In terms of potential negative societal impact, by utilizing the training and testing data from different domains, there is a possibility of data privacy concern. A potential limitation of open-vocabulary domain adaptation is that the effectiveness of open-vocabulary domain adaptation heavily relies on the quality and breadth of the pre-trained models. If these models are not sufficiently comprehensive or up-to-date, their generalization capabilities may be limited.

## G    SOURCE CODE

For the source code, please check `https://anonymous.4open.science/r/LEET-585B/README.md`.

## H    LLM USAGE STATEMENT

Large Language Models (LLMs) were used solely to aid in polishing the writing and improving the clarity of exposition. No part of the research ideation, experimental design, implementation, or analysis relied on LLMs. The authors take full responsibility for the content of this paper.

