# OpenReview forum: "Parameter-Efficient Multi-Source Domain-Adaptive Prompt Tuning for Open-Vocabulary Object Detection"
_ICLR.cc/2026/Conference — Submitted to ICLR 2026_

### Official Review · Reviewer_eVdX · 2025-10-29

**Soundness:** 3
**Presentation:** 3
**Contribution:** 2
**Rating:** 4
**Confidence:** 4

**Summary:**

This paper proposes a new method called MAP for Cross-domain open-vocabulary object detection where model is trained on multiple source domains and aims to detect novel objects in the unseen target domain. MAP includes 2 main modules: Multi-Source Prompt Learning (MSPL) and Unsupervised Target Prompt Learning (UTPL). MSPL learns both domain-specific and class specific soft prompts while UTPL aims to improve performance on novel objects in target domain through minimizing entropy.

Paper conduct comprehensive experiments on artistic and adverse-weather domain adaptation benchmarks and compare against standard prompt tuning adaptation methods, multiple Open-Vocab object detection models, and multi-source cross domain adaptation related works. Experimental results shows that the proposed method achieves state of the arts compared to benchmarks and ablation studies shows that UTPL, specifically the orthogonal loss on novel classes can significantly improve performance on novel objects while MSPL objectives mainly help with cross-domain generalization

**Strengths:**

1- Paper studies is a more challenging setting: Cross-domain open-vocabulary object detection where the model requires generalization to unseen domains while detecting novel objects. This setting better resembles practical and real-world scenarios.

2- Results are comprehensive on different types of domain shift, showcasing their effectiveness

3- Model achieves sota performance compared to all related works, including standard single-source DA methods, open-vocab models, and multi-source DA prompt tuning approaches.
4- Experimental setup is comprehensive and includes multiple architectures such as regionCLIP, YOLO-World, OVMR, and GDINO, showcasing the effectiveness and generalizability of results and findings.

**Weaknesses:**

Weaknesses

**Single Source DA baselines experimental setup is unfair**
 (1) Single-source DA methods (e.g., DAPL) are trained on only one source domain, while the proposed MAP method is trained on multiple source domains. Hence, this comparison may not showcase the effectiveness of the proposed method (i.e., MSPL and MSPL+UTPL) compared single-source baselines. Hence, its more fair to train single source domains on a *combined source domains*  (i.e. combine data from multiple sources and consider them as 1 source domain) like [1] and/or train MAP( MSPL , MSPL+UTPL) on a single source DA setup for a fair comparison.

(2) MAP uses domain-specific and class-specific prompts. How’s the prompt designed in baselines like DAPL? Many baseline methods like Coop , DAPL, etc, can naturally be expanded to use class-specific learnable prompts. For instance, DAPL already has its extension in their paper (see Eq. 6). Hence, clarifying the experimental setting for baselines and whether class-specific prompts are used could better help to evaluate the effectiveness of the proposed method. Especially because MSPL is highly similar to DAPL approach


While UTPL significantly boosts the performance on novel objects, both the prompt design (i.e., using class-specific prompts) and L_novel objective require the availability if C_novel labels during the training. While I acknowledge that the exact annotation for novel categories is not used during the training, the proposed method may be ineffective when novel objects are unknown during the training time.  I’d appreciate if this could be further clarified, as UTPL is the main distinction from prior works (i.e. not including pseudo labels in DA methods).



CLIPStyler suffers from distorting content and introducing artifacts that can impede its usability in downstream tasks like object detection and segmentation (as shown in examples Fig.3 in the appendix and [6])  can hurt the performance, especially on object detection. While I appreciate the comparison to different augmentation methods (e.g., CycleGan) in the appendix that shows relatively comparable performance, more comparison against more current diffusion-based style transfer models could be helpful, which do not have some of the limitations of traditional approaches like CycleGan and can be used off the shelf. There are many diffusion-based editing or style transfer methods, some examples: [2,3,4,5]


Some minor issues:

L_novel in Eq12 is only based on novel class prompts and base class prompts. However, in Fig.1(b) there is a connection between L_novel and target images. So the connection should be removed, or please clarify.

In Table.2, YOLO-World, for DR SAP (38.42) should be bolded.



Overall, I believe the paper is studying an important problem and the method is shown to be effective. Hence, addressing the above comments and questions can help better understand the effectiveness and contribution of the method, its contributions, and limitations. Given the rebuttal, I'm open to adjusting my score.

[1] Semantic-aware adaptive prompt learning for universal multi-source domain adaptation
[2]  Khawar Islam et al., Diffusemix: Label Preserving Data Augmentation with Diffusion Models, CVPR, 2024.
[3] Generalization by Adaptation: Diffusion-Based Domain Extension for Domain-Generalized Semantic Segmentation, Wacv24
[4] Zero-Shot Contrastive Loss for Text-Guided Diffusion Image Style Transfer, ICCV'23
[5] InstructPix2Pix: Learning to Follow Image Editing Instructions, CVPR'23
[6] PØDA: Prompt-driven Zero-shot Domain Adaptation

**Questions:**

- In UTPL target domain and its style transferred versions are used. What would be the performance if corresponding objectives are only applied to the target image? i.e., excluding the CLIPStyle augmentation module? This could better help the effectiveness of this component as it adds some computational overhead.
- How would the model perform when faced with an “unseen novel category” during inference?
- How would using a diffusion-based augmentation method impact the performance?
- [minor] Currently, the distinction of paper (especially MSPL and theoretical analysis) is vague to prior works, especially in the related work section. While theoretical analysis is still interesting even if adopted from another work, clearly explaining differences could better help readers to understand the paper and its contributions.

---

> ### Author Response · Authors · 2025-11-23
>
> **Q1. Single Source DA baselines experimental setup is unfair.**
>
> Thank you for the insightful comment. We agree that single-source domain adaptation (SSDA) and multi-source domain adaptation (MSDA) settings differ, and that directly comparing SSDA methods trained on a single source to MSDA methods is not fully aligned. In our main experiments, each single-source baseline (LB, CoOP, DAPL, DAPro) is trained following its original setting, i.e., only one source domain is used at a time, because this is how these methods are designed.
>
> However, we appreciate the reviewer’s suggestion to evaluate these baselines using all source domains combined, treating the union of sources as a single aggregated source. We conducted additional experiments accordingly, and the results are shown below:
>
>
>
> |       |  AP  | AP_base | AP_novel |
> |-------|------|---------|----------|
> | LB    | 46.14|  47.08  |  43.85   |
> | CoOP  | 45.24|  46.08  |  43.21   |
> | DAPL  | 46.54|  47.58  |  44.62   |
> | DAPro | 47.85|  48.88  |  45.49   |
>
> These results show that incorporating information from multiple source domains indeed boosts the performance of SSDA methods. This further reinforces the importance of properly integrating heterogeneous sources, something that our proposed meta-learned aggregation framework is explicitly designed to address.
>
> **Q2. Prompt designed in baselines like DAPL.**
>
> Thank you for the helpful comment. Our prompt design for baselines strictly follows the configurations described in their original papers. To ensure fair and reproducible comparisons, we use the standard domain-adaptive prompt formulations provided in their published implementations.
>
> MAP follows the same prompt design as DAPro, using [domain-specific token, domain label] and [class-specific tokens, class label], with the only addition being a mask for novel classes. DAPL employs a similar structure: [domain-specific token] and [class-specific tokens, class label], but does not include the domain label, which can be extended easily. This highlights that our prompt architecture is largely consistent with prior work, while the novelty of MAP lies in the multi-source prompt learning (MSPL) and unsupervised target prompt learning (UTPL) objectives rather than the prompt design itself,
> (1) the multi-source prompt learning (MSPL) loss that aligns prompts across heterogeneous source domains, and
> (2) the unsupervised target prompt learning (UTPL) loss that adapts prompts to the target domain without labels.
> These components are not present in DAPL, CoOP,  or DAPro, and they account for the performance improvements observed across both single-source and multi-source baselines.
>
> **Q3. Clarification of $C_{novel}$ availability.**
>
> Thank you for the constructive comment. Our setting follows the standard open-vocabulary domain adaptation formulation, where the names of the novel classes are provided but no labels or annotations from these classes are used during training. This is consistent with prior work in open-vocabulary detection, where class names (seen or unseen) define the text embeddings but supervision is available only for seen classes.
>
> UTPL does not rely on any ground-truth annotations for $C_{novel}$; it only uses the novel class names to form text embeddings, which is a common and necessary assumption in open-vocabulary models. Without class names, even text-based vision-language models (e.g., CLIP, OWL-ViT) cannot score or detect unseen categories. Importantly, the contribution of UTPL is not in assuming additional knowledge about novel classes, but in learning how to adapt prompts to the target domain without labels, ensuring that both seen and novel prompts align with the target distribution. This mechanism remains valid and effective as long as class names are available, a standard assumption in open-vocabulary detection, and does not require any labeled novel instances. We have added this prerequisite to the problem statement.

---

> ### Author Response · Authors · 2025-11-23
>
> **Q4. Comparison against diffusion-based augmentation.**
>
> Thank you for the insightful comment. We agree that CLIPStyler may introduce artifacts, and we already observe this in our appendix. Our choice of CLIPStyler employs lightweight, controllable text-driven style augmentation. We appreciate the reviewer’s suggestion to consider diffusion-based style transfer methods. We conducted additional experiments using diffusion-based augmentation [2,3,4] and PODA[5]; the results are summarized in the following table:
>
>
> |                     |  AP  | AP_base | AP_novel |
> |---------------------|------|---------|----------|
> | Diffusemix [2]      |53.54 |  54.28  |  53.08   |
> | DIDEX [3]           |48.54 |  48.99  |  47.62   |
> | ZeCon [4]           |49.54 |  49.99  |  48.62   |
> | InstructPix2Pix [5] |53.78 |  55.18  |  53.70   |
> | PODA [6]            |50.62 |  50.87  |  49.49   |
>
>
> The results show that diffusion-based augmentations produce higher-quality stylized images and generally lead to improved adaptation performance, further confirming that style-based augmentation effectively reduces the domain gap. This also demonstrates that our framework is flexible and can directly benefit from more advanced generative models as they continue to improve.
>
> **Q5. Fig.1(b) $\mathcal{L}_{novel}$.**
>
> Thank you for carefully reviewing our paper and pointing out this issue. You are correct that the connection shown in Figure 1(b) between $L_{\text{novel}}$ and the target images is misleading. In the revised figure, $L_{\text{novel}}$ is correctly shown as being connected only to the source and target prompts, and not to the target images. We have updated the figure in the revised manuscript to clarify this.
>
> **Q6. Ablation study of removing CLIPStyle augmentation module.**
>
> Thank you for the suggestion. We conducted an ablation where the CLIPStyler augmentation module is replaced with standard augmentation techniques, including random cropping, translation, and rotation, to generate multiple views of the source images, please refer to Reviewer EDfu Q2 for the result and analysis. This experiment allows us to evaluate the contribution of CLIPStyler’s text-guided style augmentation compared to conventional augmentations and assess its impact on both base and novel class performance.
>
>
> **Q7. How would the model perform when faced with an “unseen novel category” during inference?**
>
> In our framework, the model operates under an open-vocabulary setting, where novel class names are known, but no labeled examples from these classes are available during training. The learned target prompts for novel classes are adapted in an unsupervised manner using UTPL, leveraging only unlabeled target data to align the novel-class prompts with the target domain distribution.
>
> If the model encounters a truly unseen novel category, i.e., a class whose name was not provided during training, it cannot detect it reliably. This is because the domain adaptive prompt learning requires the class label or description to construct the prompt. In other words,  UTPL allows the model to generalize to novel classes with known names but no labels and it does not allow zero-shot detection of completely unknown classes whose names were never seen. This limitation is intrinsic to open-vocabulary models that rely on text embeddings. For a truly unseen novel category during inference, the model behaves like a zero-shot object detector. In this case, its performance is equivalent to that of methods such as LB, which rely solely on the pretrained model without any adaptation.
>
> **Q8. Differences of theoretical analysis to prior works.**
>
> Thank you for acknowledging the contribution of our theoretical analysis. Inspired by Wang et al. (2024a), we analyzed the behavior of supervised prompts in terms of fidelity and distinction, and we further proposed unsupervised fidelity to analyze the behavior of unsupervised prompts.
>
> Unlike prior work, which primarily focuses on practical prompt learning and selection, our theoretical analysis (in MSPL and UTPL) provides formal guarantees regarding the learning objectives and offers deeper insights into how and why prompt learning effectively aligns knowledge across multiple domains. This constitutes a distinct contribution beyond empirical prompt evaluation.

---

### Official Review · Reviewer_nBMU · 2025-10-30

**Soundness:** 3
**Presentation:** 3
**Contribution:** 2
**Rating:** 4
**Confidence:** 4

**Summary:**

This paper studies cross-domain open-vocabulary object detection (OVOD), a challenging problem that requires models to generalize across both domain shifts and category shifts. To jointly address these two types of shifts, the authors propose MAP, a parameter-efficient Multi-source domain-Adaptive Prompt tuning framework.

The proposed MAP framework consists of two core components: (1) Multi-Source Prompt Learning (MSPL), which learns both shared and domain-aware prompts from multiple source domains; (2) Unsupervised Target Prompt Learning (UTPL), which encourages prediction consistency across augmented target images.

Extensive experiments on multiple benchmark datasets demonstrate that MAP achieves state-of-the-art performance. The paper also provides theoretical analysis to support the proposed design.

**Strengths:**

The paper addresses the joint challenge of domain shifts and category shifts in cross-domain OVOD through a well-structured prompt-based learning framework. The introduction of MSPL and UTPL offers a unified approach to handle both adaptation and generalization.

Experimental results are comprehensive, covering multiple benchmarks, and the method consistently outperforms prior approaches. Theoretical justifications further enhance the credibility of the framework.

**Weaknesses:**

While the idea of UTPL contributes to the overall framework, its novelty is limited. The use of prediction consistency across augmented views as an unsupervised signal is common in both unsupervised domain adaptation (UDA) and unsupervised prompt tuning literature, as in prior works such as [a]. The main distinction here lies in the augmentation strategy, where the authors employ CLIPstyler to generate augmented samples. However, the paper does not clearly state whether these augmentations are generated dynamically during training or precomputed. If they are generated online, the additional computational cost should be explicitly discussed. Moreover, comparisons with alternative augmentation techniques are necessary to demonstrate that the CLIPstyler-based augmentation leads to superior adaptation.

[a] Test-Time Prompt Tuning for Zero-Shot Generalization in Vision-Language Models

The comparison methods is insufficient. Since the proposed work is closely related to unsupervised prompt tuning, it would be more convincing to include comparisons with recent methods in this area. Furthermore, the fairness of comparisons with few-shot prompt tuning methods such as CoOp should be clarifiet, particularly regarding how data usage and supervision levels are aligned during evaluation.

Finally, the proposed MAP framework, while effective, appears to be task-agnostic. Its design is relatively general and could potentially apply to other tasks such as classification or segmentation. However, the paper lacks components specifically tailored to object detection, such as detection-aware prompt adaptation or task-specific loss formulations. As a result, the methodological novelty for the detection task itself is somewhat limited.

**Questions:**

Please see the Weakness.

---

> ### Author Response · Authors · 2025-11-23
>
> **Q1. Limited novelty of UTPL.**
>
> We appreciate the reviewer’s observation. We fully agree that prediction consistency across augmented views has been explored in UDA and unsupervised prompt learning literature. However, our UTPL differs from prior consistency-based objectives in two important ways: (1) Existing works use consistency primarily for feature alignment or regularization. In contrast, UTPL is specifically designed to recover class-level semantic invariances across cross-domain style transformations and to stabilize prompt learning under noisy or sparse labels. Its role is not merely regularization but to explicitly guide the transformation from base-class prompts to novel-class representations in a label-limited setting. (2) Prior works typically rely on low-level or mild geometric/color augmentations. Our method introduces a semantic style-transfer augmentation via CLIPstyler, where each transformed view shifts domain-level appearance while preserving category-defining semantics. This minimizes domain gap between the source models and the unlabeled target while produce cross-domain semantic perturbations, which cannot be achieved by standard augmentations used in previous consistency frameworks. Consequently, UTPL enforces consistency over source-stylized, domain-shifted views rather than over simple view-shifted samples, ensuring that the resulting prediction consistency signal remains meaningful and informative for cross-domain prompt learning.
>
> **Q2. Augmentations are generated dynamically or precomputed.**
>
> The augmentations are precomputed before training. This design reduces computational overhead during training while still providing sufficient diversity for effective cross-domain prompt learning.
>
> **Q3. Alternative augmentation techniques.**
>
> Thank you for the suggestion. We have compared our method with traditional augmentation strategies, including random cropping, translation, rotation, and Gaussian noise,  diffusion based augmentation (please refer to reviewer eVdX, Q4) and image-to-image augmentation methods(please refer to Appendix D.2 ). The results are presented in the following table,
>
> |                        | AP    | AP_base | AP_novel |
> |------------------------|-------|---------|----------|
> | Traditonal Augmentation| 46.52 |  48.24  |   44.35  |
> | MAP                    | 55.55 |  56.67  |   55.16  |
>
>
> **Q4. Additional comparison methods and clarification.**
>
> Thank you for the suggestion. To provide a more comprehensive evaluation, we included comparisons with recent few-shot and unsupervised prompt tuning methods that are closely related to our setting in the following table:
>
> |         | AP    | AP_base | AP_novel |
> |---------|-------|---------|----------|
> | TPT[a]  | 46.37 |  47.45  |   45.52  |
> | UPT[b]  | 44.33 |  44.95  |   43.61  |
> |M-CRPL[c]| 40.28 |  41.35  |   38.84  |
> | MAP     | 55.55 |  56.67  |   55.16  |
>
>
> These approaches represent strong and up-to-date baselines in unsupervised prompt tuning. Their performance is generally in line with other single-source prompt learning approaches. In contrast, our method explicitly leverages multiple heterogeneous source domains and integrates them through the meta-learned aggregation mechanism, which leads to substantially better performance across base and novel categories.
>
>
> [b] He, Weizhen, et al. "Unsupervised prompt tuning for text-driven object detection." ICCV23.
>
> [c] Vuong, Tung-Long, et al. "Preserving Clusters in Prompt Learning for Unsupervised Domain Adaptation." CVPR25.
>
>
> **Q5. Supervision setting for comparison baselines.**
>
> In particular, our setting uses no labeled target data, and therefore baselines that require few-shot labels are restricted to using the same amount of source-domain supervision during training, without access to target labels. This ensures a fair comparison between our unsupervised target-prompt learning and methods that rely on supervised prompt tuning.

---

> ### Author Response · Authors · 2025-11-23
>
> **Q6. Performance of MAP on classification or segmentation tasks.**
>
> Thank you for the insightful comment. We agree that MAP is designed as a task-agnostic framework. This is intentional: our formulation focuses on domain-robust prompt learning and emphasizes generality rather than injecting task-specific heuristics. We chose to primarily evaluate on object detection because (1) detection is substantially more complex than classification, requiring localization + classification jointly, and (2) it has broad real-world relevance in applications such as autonomous driving, robotics, and surveillance systems. Demonstrating strong results on such a challenging and practical task shows the robustness and applicability of MAP.
>
> To further support the generality of the framework, we additionally conduct experiments on classification (DomainNet) and semantic segmentation (ACDC), confirming that the same MAP formulation transfers well to other tasks without architectural changes. For both tasks, we follow the open-vocabulary protocol by removing a subset of classes from the source domain to ensure no label overlap with the target domain.
>
> |           |  AP  | AP_base | AP_novel |
> |-----------|------|---------|----------|
> | Clipart   | 92.88|  94.45  |   91.52  |
> | Infograph | 90.67|  91.23  |   90.12  |
> | Painting  | 88.45|  89.65  |   88.01  |
> | Quickdraw | 89.98|  88.79  |   90.24  |
> | Real      | 87.65|  88.42  |   86.98  |
> | Sketch    | 91.23|  91.88  |   90.47  |
>
> |           |  AP  | AP_base | AP_novel |
> |-----------|------|---------|----------|
> | Foggy     | 33.7 |   34.8  |   30.5   |
> | Nighttime | 33.1 |   33.9  |   30.4   |
> | Rainy     | 31.5 |   32.4  |   28.2   |
> | Snowy     | 31.2 |   32.2  |   28.1   |
>
>
>
> While the reviewer notes that MAP does not introduce detection-specific prompt components or detection-aware losses, the goal of our method is not to redesign the detection head. Instead, MAP provides a unified multi-domain prompt learning mechanism that can be plugged into any existing vision-language model across tasks. This generality is a key strength: MAP improves domain adaptation performance in object detection without requiring task-specific engineering, and simultaneously extends to other tasks with minimal modifications.

---

### Official Review · Reviewer_FVtF · 2025-10-31

**Soundness:** 3
**Presentation:** 3
**Contribution:** 3
**Rating:** 4
**Confidence:** 4

**Summary:**

This paper proposes a training framework to open-vocabulary object detection, and aims to address the issue of distribution shift (domain and semantic shift) in this area. The authors introduce theoretical analysis for the proposed method. Extensive experiments are conducted.

**Strengths:**

The proposed method seems reseaonable.

Extensive experiments are conducted.

Solving distribution shift is essential in open-vocalbulary tasks.

It is good to see the authors considering both semantic and domain shift.

**Weaknesses:**

Can you please clarify, in Figure 2, which object is the novel one? It is hard to tell the performance on novel classes from the quality results.

UnFI is not defined in Equatin 15.

In the theoretical part, I think the Propositions 4.1, 4.2 and 4.3 are more like assumptions or definitions. And for 4.3, the authors mentioned that $\hat{y}$ is predicted, which means $H(\hat{y})$ is not a constant. So, only minimizing the conditional entropy may not increase the mutual information. I mean, for example, if the model weights are all 0, the conditional entropy is 0 but the mutual information is 0 too, or please correct me if I am wrong.

For the proof in 4.4, the Equation 21 is concluded from Equation 20, but the $y$ in the two equations are different. Can you please clarify? and based on this, can you please clarify the theory followed by that?

Can you please clarify why the mask tokens can represent novel classes? And how many tokens are used for this and why?

minor typos: leaning-> learning, $\mathcal{L}$ novel-> $\mathcal{L}_{novel}$ (line 439)

**Questions:**

Please see above weaknesses.

---

> ### Author Response · Authors · 2025-11-23
>
> **Q1. Novel objects in Figure 2.**
>
> Thank you for pointing this out. The example in Figure 2 does not include any novel classes. In the DWD dataset, the novel categories are traffic light and train, which are not present in the current qualitative example. We've updated the paper to include an additional qualitative figure that explicitly shows detections on novel classes. The figure shows that our method successfully detects novel class such as Traffic light.
>
>
> **Q2. UnFI is not defined in Equation 15.**
>
> Thank you for pointing this out. UnFI refers to the Unsupervised Fidelity term, which measures the alignment quality of the target-domain prompt without requiring target labels. We have added its full definition and description in the revised manuscript to ensure clarity.
>
> **Q3. Propositions 4.1, 4.2 and 4.3.**
>
> Thank you for the thoughtful comment. We agree with the reviewer’s observation and have revised Propositions 4.1–4.3 as Definitions. For 4.3, given the transferable knowledge from the source to the target domain, we've made a non-degenerate label distribution assumption under the domain adaptation setting, a standard and reasonable assumption, where the target predictions are expected to vary meaningfully with input. Moreover, our source-stylized augmentation of target images further reinforces this assumption by reducing the domain gap between the source models and the target representations, ensuring that the prediction distribution remains informative and non-degenerate. Under this assumption, the collapse scenario described by the reviewer, where minimizing conditional entropy leads to zero mutual information, is already excluded. Consequently, minimizing conditional entropy encourages confident, input-dependent predictions, and the situation described by the reviewer will not occur.
>
>
> **Q4. Proof of 4.4.**
>
> Thank you for your thoughtful comment. In Equation (20),  $y$ denotes the ground-truth class label, whereas in Equation (21), $\hat{y}$ denotes the predicted class distribution (or logits) output by the model. For Equation (20), the relationship $\text{Min} H(y_k|\mathbf{pt}_k) \Rightarrow \text{Max } \text{MI}(\mathbf{pt}_k; y_k)$ holds directly,since $H(y_k)$ is constant. For Equation (21), we rely on the non-degenerate label distribution assumption introduced in Proposition 4.3. Under this assumption, the prediction distribution $\hat{y}$ is non-constant and varies meaningfully with the input, which guarantees that  $H(\hat{y})>0$. This prevents model collapse and ensures that minimizing $H(\hat{y}|\mathbf{x})$ indeed increase $MI(\mathbf{x};\hat{y})$. We have revised the proof accordingly to clarify how Equation (21) follows under this assumption.
>
>
> **Q5. The use of mask.**
>
> The learnable mask tokens serve as class-agnostic placeholders that specialize toward novel classes during adaptation. Because these tokens are optimized jointly with the orthogonality loss, they are encouraged to diverge from the base-class prompt features. This enforced orthogonality prevents the novel-class prompts from collapsing toward the well-learned base-class representations. As a result, the model avoids overfitting to base classes and maintains the flexibility needed to represent novel-category semantics effectively. In Appendix D.3, we conduct an ablation study on $M_3$,the number of mask tokens. The results show that performance improves as $M_3$ increases up to a certain point, and the optimal performance is achieved at $M_3 = 9$. Therefore, we adopt $M_3 = 9$ in all experiments.
>
> **Q6. Minor typos**
>
> Thank you for carefully reading our paper and pointing out the typos. We have corrected all of them in the revised version.

---

### Official Review · Reviewer_EDfu · 2025-11-02

**Soundness:** 3
**Presentation:** 3
**Contribution:** 3
**Rating:** 6
**Confidence:** 5

**Summary:**

This paper targets Cross-domain Open-Vocabulary Detection (COVD) and proposes MAP, a parameter-efficient framework that unifies Multi-Source Prompt Learning (MSPL) with Unsupervised Target Prompt Learning (UTPL): MSPL learns class-aware and domain-aware prompts, while UTPL adapts to unlabeled targets via text-guided style augmentations and an entropy-minimization consistency loss, avoiding pseudo-labels.

**Strengths:**

It further introduces a novel-class mask and mutual-orthogonality regularization, and analyzes prompt fidelity and distinction. Experiments report state-of-the-art COVD results with minimal extra parameters (backbones frozen; only prompts optimized).

**Weaknesses:**

[1] Clarify assumptions：Say whether the method needs prior knowledge of novel classes. State which prompts are used at inference. Please explain the selection rule.

[2] Run three small ablations：First remove text guided style augmentation. Second remove entropy minimization and consistency. Third replace hand crafted domain descriptions with automatic or generic ones. Report the change in AP for each test.

[3] Do the gains hold under mAP@[.5:.95]? Were all baselines run with the same single or multi source setting and the same backbone?

**Questions:**

See Weaknesses.

---

> ### Author Response · Authors · 2025-11-23
>
> **Q1.Clarify assumptions.**
>
> We thank the reviewer for the suggestion. Our method does not require prior knowledge of the novel classes. During training, only base-class labels are used; prompts corresponding to novel classes are not introduced until inference time. During inference, the model uses the learned target-domain prompt, i.e., $\textbf{pt}^t$.
>
> **Q2. Ablations.**
>
> Thank you for the helpful suggestions. We performed all three ablations requested by the reviewer. The AP values for each ablation are summarized below, note that for multi-source domain adaptation, we use Watercolor2k and Comic2k as sources and evaluated on Clipart1k with RegionCLIP backbone as our test platform:
>
> - (1) Removing text-guided style augmentation.
> We replace the text-guided source-style augmentation with generic augmentations (random cropping, translation, rotation, Gaussian noise). Because these augmentations do not preserve source-domain style, the domain gap between the source model and the augmented views increases, leading to a noticeable performance drop compared to using text-guided style augmentation.
>
> - (2) Removing entropy minimization and consistency.
> We remove the target-domain losses includes $L_{\text{class}}^t$, and $L_{\text{domain}}^t$, and train UTPL only with $L_{\text{novel}}$ $L_{\text{orth}}$. Without entropy minimization and consistency, the target prompt cannot learn meaningful class-specific or domain-specific tokens. As a result, the remaining regularization terms are insufficient to guide the target prompt, and performance degrades significantly.
>
> - (3) Replacing domain-specific descriptions with generic ones.
> Previously we use domain-specific descriptions such as “comic’’ (comic2k) or “watercolor’’ (Watercolor2k). We replace them with generic descriptions: “art’’. These generic descriptions encode little domain information, so the target prompt is less domain-aware, which reduces adaptation performance.
>
>
>
> |                                             |    AP   | AP_base | AP_novel |
> |---------------------------------------------|---------|---------|----------|
> | remove text guided style augmentation       |  46.52  |  48.24  |   44.35  |
> | remove entropy minimization and consistency |  41.88  |  42.63  |   39.59  |
> | generic domain descriptions                 |  50.92  |  49.77  |   48.65  |
>
>
> We have added this ablation studies in the revised paper.
>
> **Q3. mAP@[.5:.95] and baseline settings**
>
> Thank you for the insightful comment. Our main paper reports AP@.5; here we additionally include AP@[.5:.95] for the GDINO backbone on Clipart1k. As shown below, the performance gain of our method is preserved under the stricter COCO-style metric. We have added AP@[.5:.95] for all datasets and backbones in the revised version.
>
> |       | AP      | AP_base | AP_novel |
> |-------|---------|---------|----------|
> | LB    |  35.47  |  36.52  |  33.48   |
> | CoOP  |  35.67  |  36.78  |  33.77   |
> | DAPL  |  35.96  |  37.21  |  34.18   |
> | DAPro |  36.84  |  38.12  |  35.05   |
> | MPA   |  39.52  |  40.68  |  37.65   |
> | SAP   |  40.62  |  41.79  |  38.73   |
> | APNE  |  42.85  |  43.54  |  41.29   |
> | POND  |  46.17  |  46.95  |  45.08   |
> | MAP   |  50.29  |  50.88  |  48.79   |
>
>
>
> Regarding baseline settings, all baselines are run with the same backbone to ensure a fair comparison. The methods differ only in whether they assume a single-source or multi-source setting, following the original papers:
> (1) Single-source baselines: LB, CoOP, DAPL, DAPro; (2) Multi-source baselines: MPA, SAP, APNE, POND. We have clarified this in the revised manuscript.

---

### Author Response · Authors · 2025-11-23
**Summary of Changes**

To address the reviewers’ comments, we have provided a detailed rebuttal for each individual review. In addition, we have revised the paper accordingly to incorporate these changes. Main revisions fall into the three main aspects:
(1) Additional Experiments, (2) Clarification and Strengthening of Theoretical Analysis, and
(3) Improvements to Presentation and Writing.
Detailed changes under each category are listed below.

- **Additional Experiments**
    - Added ablations on text-guided style augmentation, entropy minimization, consistency regularization, and domain-specific descriptions.
    - Included additional metric AP@[.5:.95] for more comprehensive evaluation.
    - Added comparisons with alternative augmentation techniques, including diffusion-based methods (DiffuseMix, DIDEX, ZeCon, InstructPix2Pix) and PODA.
    - Added comparisons with recent few-shot and unsupervised prompt-tuning methods.
    - Added evaluation on additional tasks: classification (DomainNet) and segmentation (BDD100K).
    - Added experiments where all single-source DA baselines are trained on the combined source domains for fairer comparison.

- **Clarification and Strengthening of Theoretical Analysis**

    - Renamed Proposition 4.1, 4.2, 4.3 to Definition 4.1, 4.2, 4.3 for accuracy.
    - Defined UnFI before its first appearance to improve clarity.

- **Improvements to Presentation**
    - Updated Figure 1(b) for improved visual clarity.
    - Replaced Figure 2 to include novel-class detection results.
    - Fixed various typos and improved overall writing quality.

All the changes are highlighted in blue so that they can be easily identified by the reviewers.

---

### Meta-Review · Area_Chair_bNbo · 2026-01-03

**Summary:**

The submission initially received relatively negative reviews. The main concerns can be summarized as follows:
1. The technical and methodological novelty is limited.
2. The additional computational cost should be explicitly discussed.
3. The comparisons seem unfair.
4. More implementation details and ablations are needed.

In the rebuttal, the authors have provided additional experimental results and ablations. In my opinion, the main concerns about technical novelty still remain. Thus, I recommend Reject.

**Reviewer Concerns:**

The concerns about more implementation details and ablations are addressed, but some concerns about the novelty still remain.

**Reviewer Scores:**

I think that all reviewers will not change their ratings.

---

### Decision · Program_Chairs · 2026-01-26

Reject